



# Recent Evolution of Supraglacial Lakes on ice shelves in Dronning Maud Land, East Antarctica

Anirudha Mahagaonkar[1,2], Geir Moholdt[1], Thomas V. Schuler[2]

[1]Norwegian Polar Institute, Tromsø, 9296, Norway

[2]Department of Geosciences, University of Oslo, Oslo, 0315, Norway

*Correspondence to*: anirudha.mahagaonkar@npolar.no

## Abstract

Accumulation of meltwater on the surface of ice shelves can have severe impacts on the ice sheet – ice shelf stability regime. Meltwater ponding on ice shelves can cause firn air depletion, flexure, hydrofracture and collapse of shelves that could further

lead to increased ice-sheet discharge and sea level rise. Despite recent progress in the mapping of supraglacial lakes around Antarctica, there is still limited understanding of their dynamics, climatic and environmental controls, and their role in the Antarctic ice sheet mass budget. In this study, we track the seasonal and interannual evolution of supraglacial lakes on ice shelves in Dronning Maud Land in East Antarctica. The assessment employs an automatized band-thresholding approach to examine nearly 2500 Landsat-8 and Sentinel-2 scenes captured between November and March from 2014 to 2021. Large

networks of supraglacial lakes and streams were identified over the ice shelves Riiser Larsen, Nivlisen and Roi Baudouin, whereas other ice shelves had only smaller areas with isolated ponds (Fimbulisen and Muninisen) or no significant meltwater lakes at all. Despite large interannual variations in surface ponding, in specific melt years the relative extents were mostly consistent between different ice-shelf regions. The peak extents of supraglacial lakes typically occurred in mid-to-late January and varied from $38.19 \pm 29.53$ km$^2$ in the low melt-year 2020-2021 to $809.37 \pm 206.74$ km$^2$ in the high melt-year $2016 – 2017$,

corresponding to water volumes of $0.03 \pm 0.02$ km$^3$ and $0.67 \pm 0.16$ km$^3$ respectively. Comparison with positive degree days and seasonal temperatures shows considerable correlation with maximum lake extent for some ice shelves, but in total, it cannot explain the large differences in magnitudes of surface ponding over different ice shelves. For instance, melt extents of Fimbulisen and Nivlisen that lie next to each other differ by two orders of magnitude. Our assessments provide important insights into surface hydrology over the region and will be helpful to further constrain the different processes that control the

evolution of supraglacial meltwater systems in Dronning Maud Land.

## 1 Introduction

The response of the Antarctic Ice Sheet to climate change and its contribution to sea level rise is closely related to changes that the associated ice shelves undergo (Dupont and Alley, 2005; Fürst et al., 2016; Gagliardini et al., 2010; Pritchard et al., 2009; Rignot et al., 2013). Although meltwater runoff is a minor component in the mass balance of the ice sheet (Bell et al., 2018;

van Wessem et al., 2018), there is substantial surface melting in many coastal regions (Trusel et al., 2012) and also widespread



presence of supraglacial lakes and streams (SGL). These lakes form due to melting of snow and ice (Echelmeyer et al., 1991) and subsequent ponding of meltwater in topographical depressions, typically seen around grounding zones (Stokes et al., 2019). Accumulation of meltwater in SGLs may lead to flexure of ice shelves (Banwell et al., 2013, 2019), hydro-fracture (Scambos et al., 2000; Rott et al., 1996), and collapse (DeConto and Pollard, 2016), as seen in the case of Larsen-B ice shelf (Banwell et al., 2013), thereby destabilizing the ice sheet upstream due to decreased buttressing. In other cases, meltwater saturates the underlying firn layer (Leeson et al., 2020; Lenaerts et al., 2016) as refrozen ice or firn aquifers (Kuipers Munneke et al., 2014), or in rare cases it drains subglacially and may cause glacier speedup as observed in the Antarctic Peninsula (Tuckett et al., 2019). These consequences make SGLs a key component of the ice sheet - ice shelf stability regime and highlight the need to monitor their development in times of climate change. Pervasive meltwater lakes have been widely reported from Greenland (Box and Ski, 2007; Leeson et al., 2015; Liang et al., 2012; Sneed and Hamilton, 2007) and from the Antarctic Peninsula (Scambos et al., 2000), but it is only recently that the occurrence and evolution of SGLs around the fringes of East Antarctica has caught the attention of the scientific community (Kingslake et al., 2017; Stokes et al., 2019).

Stokes et al. (2019) identified widespread presence of SGLs along the coastal margin of East Antarctica, which form an active part of the surface meltwater network comprising of lateral streams and circular lakes that are actively involved in transportation of meltwater onto and across the ice shelves (Dell et al., 2020; Kingslake et al., 2017). Among the major ice shelves on the East Antarctic ice sheet, the evolution of supraglacial lakes has been observed at the Amery (Tuckett et al., 2021) and Shackleton ice shelves (Arthur et al., 2020a). However, little is known about the spatio-temporal evolution, and its environmental/climatic controls for Dronning Maud Land in East Antarctica. Given the presence of major ice shelves in the region, and the constantly evolving supraglacial meltwater network, it is important to analyze their formation and factors responsible for their evolution over seasonal and interannual scales.

Field observations and assessments of supraglacial lakes on the ice shelves of Dronning Maud Land is very limited, with only some sporadic records from Fimbulisen (Winther et al., 1996) and Roi Baudouin (Lenaerts et al., 2016). However, the advent of remote sensing and availability of multiple satellite products have made it simpler to continuously monitor these systems. Although Synthetic Aperture Radar (SAR) products have the advantage of not being impacted by clouds which are widely present over coastal Dronning Maud Land, optical products are preferred as they have a higher confidence and allow estimation of water depths (Pope et al., 2016; Sneed and Hamilton, 2007) which is an important factor for hydrological analyses. The simple configuration of optical data allows combining products from multiple sensors, like Landsat 8's Operational Land Imager and Sentinel-2's Multispectral Instrument, thereby enhancing the spatiotemporal coverage of SGLs.

In this study, we exploit the synergy of Landsat-8 and Sentinel-2 to assess the networks of SGLs in Dronning Maud Land and study their evolution between 2014 and 2021. The objectives of the study are (1) to identify areas that have supraglacial lakes, (2) to understand seasonal and interannual evolution of the supraglacial lakes over the identified areas between 2014 and 2021



and (3) to identify the environmental and climatic factors that control their development. We use a band-reflectance thresholding method (Moussavi et al., 2020; Williamson et al., 2018) and a physically based radiative transfer model (Pope et al., 2016) to map SGL extents and derive their depths, respectively. For climatic assessments, we use outputs from the European Center for Medium-Range Weather Forecasts ERA5 reanalysis (ERA5: Fifth generation of ECMWF atmospheric reanalyses of the global climate). From this work we look to bridge the knowledge gap surrounding the status of surface

meltwater lakes and streams in Dronning Maud Land, which may have implications for regional ice shelf stability.

## 2 Study Area

Dronning Maud Land (between 20ºW and 45ºE) houses some of the major ice shelves of East Antarctica. With a coastal margin of ~2600 kms and a narrow continental shelf, the ice shelves are exposed to strong ocean and sea-ice dynamics, at the same time as they have many anchor points in form of grounded ice rises and embayments formed by promontories (Goel et al.,

2020). The rugged ice topography gives rise to complex patterns of snow erosion and accumulation, including extensive blue ice areas (Figure 1). Surface melting in combination with blue-ice or shallow snow/firn packs has led to formation of several SGL systems across the periphery (Bell et al., 2018; Kingslake et al., 2017; Trusel et al., 2013, 2012). Satellite-radar backscatter time series have been used to estimate surface meltwater fluxes (Trusel et al., 2013) over the entire continent, and indicate that in Dronning Maud Land the highest surface melt flux rates are found on Roi Baudouin, at about ~120 mm w.e.yr$^{-1}$ followed

by Nivlisen ~80 mm w.e.yr$^{-1}$ and Riiser Larsen ~70 mm w.e.yr$^{-1}$ (Trusel et al., 2013). The ice-shelves of Dronning Maud Land are fed by a number of outlet glaciers, among which are Jutulstraumen, Einsetebreen (also known as Tussebreen), Potsdam Glacier and the West and East Ragnhild Glaciers. Within Dronning Maud Land, exposed blue ice is concentrated mostly to the south and southeast of the ice shelves of the region, typically on the slopes behind the grounding lines. These blue ice areas are formed from sustained scouring from the katabatic winds blowing downslope from the ice-sheet interior.

With lower surface albedo, these areas experience enhanced shortwave radiation absorption and thereby promoting surface melting and associated meltwater drainage and subsequent ponding. Antarctic-wide studies have identified SGLs on the ice shelves Riiser Larsen, Nivlisen and Roi Baudouin (Arthur et al., 2022; Moussavi et al., 2020; Stokes et al., 2019) in Dronning Maud Land, however there is a lack of finer scale intra-seasonal assessments, except for the study by Dell et al. (2020) where seasonal assessment of supraglacial lakes and streams was done for the year 2016-2017 over Nivlisen.





## 3 Data & Methods

### 3.1 Satellite Data & Acquisition

For generating lake extents and estimating lake depths, we used satellite optical scenes captured by Landsat-8 (Sensors – Operational Land Imager (OLI) and Thermal Infrared Sensor (TIRS)) and Sentinel-2 (Sensor – Multispectral Instrument (MSI)). The assessment period from 2014 to 2021 was chosen in accordance with Landsat-8 availability, whereas Sentinel-2 products are only available since 2016. All Landsat-8 and Sentinel-2 images from each austral melt season (typically from 01-Nov to 31-Mar) were manually inspected for usability and then downloaded and processed for generating the lake extent masks and lake depth grids. The inspection of images involved visual confirmation of absence of clouds over the area of interest. We did not use the cloud detection and filtration algorithm output of USGS Earth Explorer (for Landsat-8) or Copernicus Open Access Hub (for Sentinel-2) as these algorithms work based on the presence of clouds anywhere in the scene, and not excluding/including specific locations like our areas of interest. By visual inspection, we were able to include scenes that had extensive cloud cover, but still some clear areas over SGLs. Of the ~2500 scenes that were inspected, ~1100 scenes were downloaded and processed for the study.

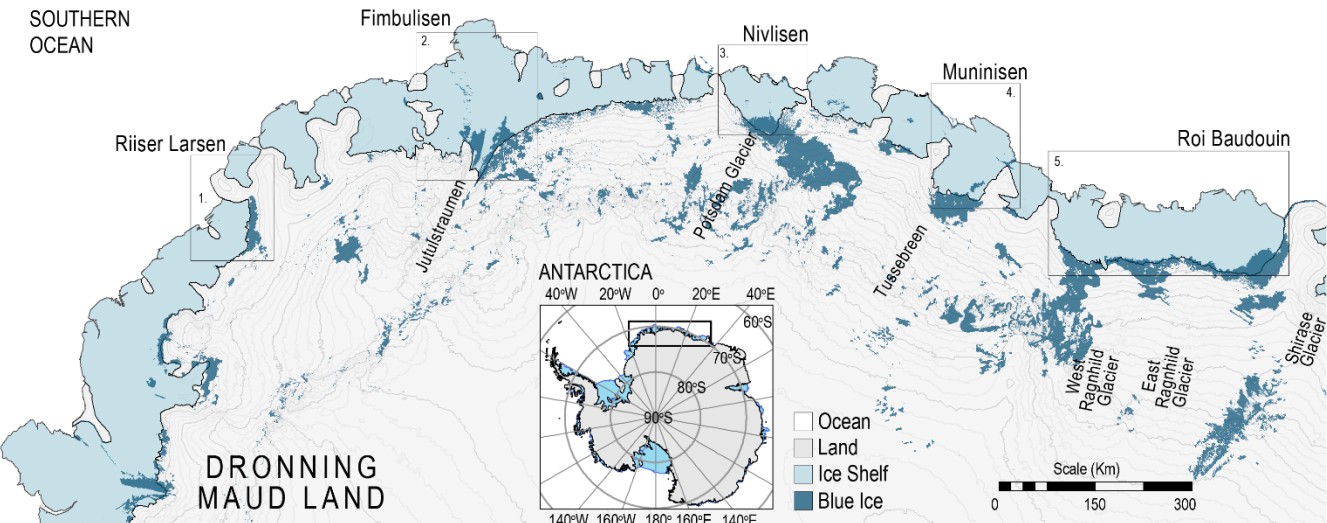

**Figure 1. Map of the Dronning Maud Land region highlighting the different ice shelves on which SGLs are identified in the study; 1. Riiser Larsen, 2. Fimbulisen, 3. Nivlisen, 4. Muninisen and 5. Roi Baudouin. The small inset map shows the location of Dronning Maud Land within Antarctica. In the background are 100 m contours representing the overall surface topography, generated using REMA 100m product (Howat et al., 2019). The dark blue coloured areas show the exposed blue ice areas as mapped from satellite imagery (Hui et al. 2004). Outlines of Ice shelves were sourced from Quantarctica.**



### 3.2 Preprocessing

Preprocessing involves conversion of raw band values to top of atmosphere reflectance and resampling of bands to match a
common pixel size independent of image source. It also involves steps for identification of features that could be sources of
misclassification.

### 3.2.1 Landsat-8

Landsat-8 scenes were downloaded from the Collection 2 Level-1 archive that provides geometrically and radiometrically
enhanced products (Landsat Collection 2 Level-1 Data, 2022). Among the 11 available spectral bands, Band 2 (corresponding
to Blue wavelength in the visible spectrum with 30m spatial resolution), Band 3 (Green – 30m), Band 4 (Red – 30m), Band 6
(shortwave infrared, SWIR – 30m) and Band 10 (thermal infrared, TIR – 100m provided as 30m) were used in the study. Pixel
values in each of the bands were first converted from raw digital numbers (DN) to top of atmosphere (TOA) reflectance using
the pixel-by-pixel sun illumination angle information from the ANG.txt file and the additive and multiplicative factors
available with the image metadata. Normalized Difference Snow Index (NDSI), which is widely used for cloud detection, was
generated using Green and SWIR bands (Hall et al., 1995). Additionally, different masks were generated, following
recommendations from Moussavi et al. (2020), to eliminate clouds, exposed rock outcrops and sea water pixels which have
been found to be major sources of misclassification (Moussavi et al., 2020; Dell et al., 2020; Williamson et al., 2018). The
cloud mask was produced by applying thresholds on NDSI and SWIR Band, whereas the rock and seawater mask was produced
using thresholds applied on brightness temperature, blue and red bands. Brightness temperature, calculated using the TIR band
(Using the USGS Landsat Level-1 Data Product, 2022), can be valuable for differentiating sunlit rocks faces and shaded rock
areas based on the difference in temperature with surrounding snow and ice. All pixels identified as clouds, rocks or sea water
were used at a later stage to filter the misclassified pixels.

### 3.2.2 Sentinel-2

Sentinel-2 products are available with radiometric and geometric corrections as Level-1C TOA reflectance data directly. Band
2 (Blue – 10m), Band 3 (Green – 10m), Band 4 (Red – 10m), Band 10 (SWIR Cirrus – 60m) and Band 11 (SWIR2 – 20m)
were used in the further processing. Band 10 and Band 11 were resampled, using bilinear interpolation, to 10 m pixel size to
match the pixel configuration of other bands. As in case of Landsat-8, we followed recommendations from Moussavi et al.,
(2020) to generate cloud, rock, and seawater masks for Sentinel-2 products. In absence of TIR band in Sentinel-2 products,
rock and seawater mask was generated using NDSI and Blue band, as against TIR and Blue band in Landsat-8. Cloud mask
was produced by applying thresholds on SWIR Cirrus and SWIR2 bands. These grid masks were used to eliminate respective
pixels from the classified output to minimize misclassification errors.





### 3.3 Delineation of SGLs

In total 1072 Landsat-8 and Sentinel-2 scenes were identified as suitable for further SGL processing that involved mapping water pixels and estimating their depths. Pixels representing SGLs were derived by applying thresholds on the Normalized
Difference Water Index $_{ice}$ (NDWI$_{ice}$) (Gao, 1996; Yang and Smith, 2013) which can identify presence of liquid water areas using the spectral characteristics from the red and blue band reflectance. NDWI$_{ice}$ is calculated following Eq. (1), where Blue and Red are the TOA values of respective bands used in estimation of the index.

$$NDWI_{ice} = \frac{Blue\ Band - Red\ Band}{Blue\ Band + Red\ Band} \tag{1}$$

Yang & Smith (2013) found that NDWI$_{ice}$ performed better in identifying water over glaciated areas than the traditional NDWI
which is more suitable for water on land. We used a threshold of 0.25 (NDWI$_{ice}$ > 0.25) to map deeper lakes (Arthur et al., 2020a; Dell et al., 2020; Williamson et al., 2018), and exclude shallow lakes (having NDWI$_{ice}$ of $0.19 - 0.25$) which otherwise would lead to overestimation of SGLs (Arthur et al., 2020a; Dell et al., 2020). This threshold was effective in detecting similar extents for both Landsat-8 and Sentinel-2 (Sect. 3.5). The binary output from the thresholding was filtered using the cloud, rock and seawater masks, eliminating the misclassified pixels. Further, we identified all connected groups of pixels and
eliminated groups that were <= 2 pixels (i.e., <= 1800 m$^2$) in size, as they would represent slush (Pope et al., 2016) or other outliers. In Sentinel-2 outputs this is equivalent to <= 18 pixels (i.e., <= 1800 m$^2$) due to its finer spatial resolution. Manual inspection was required after this step as there were still certain misclassified areas, especially when there were cloud shadows over blue ice and crevassed regions. Such areas were identified at this stage and eliminated from further processing manually. Due to the ambiguity between meltwater streams and elongated meltwater lakes, we do not classify the identified supraglacial
features into lakes and streams and collectively refer them as supraglacial lakes or SGLs.

### 3.4 Estimation of Lake Depths and Volumes

For estimating lake depths, we employed the physically based radiative transfer model (Pope et al., 2016) which is based on the attenuation of light relative to depth in the water column (Pope et al., 2016; Sneed and Hamilton, 2007). Pope et al. (2016) pointed out three assumptions for the validity of this technique – 1) there is no suspended or dissolved matter in the water
column, 2) the water surface is not severely impacted by waves or ripples, and 3) the bottom of the water body is homogenous, having similar slopes and albedo. These are reasonable assumptions for the relatively clean and homogeneous looking lakes in our study area, and the technique has been widely used for estimating depths of East Antarctic SGLs previously (Arthur et al., 2020a; Dell et al., 2020; Tuckett et al., 2021). Several studies have used the red and panchromatic bands for this estimation, doing separate calculations and reporting the mean value as the final lake depth (e.g. Moussavi et al., 2020; Williamson et al.,
2018). Since Sentinel-2 has no panchromatic band, we used only the red band to estimate lake depths for the Landsat-8 and Sentinel-2 datasets, thereby ensuring consistency in the technique. As the red wavelength from the visible spectrum is strongly


attenuated in a water column, it allows for calculating measurable changes in even small variations of depth (Box and Ski, 2007).

For pixels representing water, depth ($z$) was estimated using the rate of light attenuation ($g$), lake-bottom albedo ($A_d$), reflectance of the pixel ($R_{pix}$) relative to reflectance of optically deep water ($R_\infty$) (Philpot, 1989) in the following Eq. (2):

$$z = \frac{[\ln(A_d - R_\infty) - \ln(R_{pix} - R_\infty)]}{g} \tag{2}$$

The lake-bottom albedo ($A_d$) cannot be directly measured, so it was estimated feature-by-feature using the average red-band reflectance from the edge pixels identified by a 2-pixel buffer surrounding the respective water body. We assume that the reflectance from the lake bed is similar to the reflectance from the edges around the lake as their values would be comparable

if the lake bed was at the surface (Pope et al., 2016). The reflectance of optically deep water ($R_\infty$) can be approximated by the reflectance values from the open ocean water (Williamson et al., 2018). However, since most of our Landsat-8 and Sentinel-2 scenes do not have pixels representing open ocean, we used zero in our calculations following Arthur et al. (2020b), Banwell et al. (2019) and MacDonald et al. (2018). The depth difference between these two choices of value for $R_\infty$ has been found to be smaller than 10% (Arthur et al., 2020a). The reference pixel reflectance ($R_{pix}$) was extracted pixel-by-pixel from the red

band TOA reflectance. The rate of light attenuation (g) is slightly different for Landsat-8 and Sentinel-2 scenes due to their different wavelengths in the red band. While we used a g value of 0.7507 for Landsat-8 (Pope et al., 2016), we used 0.8304 for Sentinel-2 (Williamson et al., 2018). We found close agreement in depth estimated using the red bands of Landsat-8 and Sentinel-2 captured on the same day over the same region (Sect. 3.5). Subsequently, lake volumes were calculated by multiplying the depths with pixel area (900 m$^2$ for Landsat-8 and 100m$^2$ for Sentinel-2). After deriving lake depths for all

scenes, the Sentinel-2 outputs (area and depth) were resampled from 10 m to 30 m spatial resolution and reprojected to WGS 84 / Antarctic Polar Stereographic - EPSG:3031 projection to be consistent with Landsat-8. These time series of SGL area and volume were then used to assess seasonal evolution for each austral summer in each ice-shelf region. For interannual statistics, we only used the seasonal peak in area or volume considering the limited temporal coverage of data in certain melt seasons.

### 3.5 Cross-validation of area and depths from Landsat-8 and Sentinel-2

For using Landsat-8 and Sentinel-2 products together in a time-series, we had to ensure that the generated lake areas and volumes are consistent. To validate this, we identified four image pairs of Landsat-8 and Sentinel-2 (listed in Table S1) that were captured on the same day of the melt season and compared the lake areas and depths in the overlapping regions. We found that 3% - 5% of the derived water pixels from Sentinel-2 were classified as non-water in Landsat-8. These pixels were typically located around the lake edges or in highly inhomogeneous terrain, or they were completely isolated. This may be

attributed to the finer resolution of Sentinel-2 (10 m) than Landsat-8 (30 m), making it easier to detect small water bodies and differentiate shallow and deep lake edges. Larger areas having such pixels were manually removed, whereas smaller ones were eliminated during the feature-size-based filtration (for Sentinel-2, pixel groups smaller than <= 18 pixels were eliminated). For





depth validation, we first identified common water pixels for Landsat-8 and Sentinel-2 and then plotted their depths against each other in a scatter plot (Figure S1). The depths estimated using Landsat-8 and Sentinel-2 are highly consistent with an average $r = 0.89$ and RMSE $= 0.21$m (assessed using four pairs of images from the same dates). Despite the different configurations of the sensors, there is good agreement between lake area and volumes derived from Landsat-8 and Sentinel-2 products, validating their combined use in time-series.

**3.6 Comparison with Climatic Variables**

To assess the role of near-surface conditions on melting and ponding, we compared melt extents with near-surface air temperature, measured 2 m above the ground. Near surface air temperature was extracted from ERA5 climate reanalysis dataset at hourly resolution and then averaged to daily average temperature. We used data from single ERA5 grid points that were at similar elevations, typically near the grounding line, close to the origin of the melt features that were mapped over different ice shelves in the study. The available ERA5 grid points and the chosen grid points are shown in Figure S2. The comparison was made by calculating correlation (using the Pearson's correlation coefficient, $r$) between the December-January-February (DJF) mean temperatures, sum of positive degree days (PDD) in DJF and maximum melt area and melt volumes for each study region and each melt seasons between 2014 and 2021. The PDD was calculated as the sum of all positive temperatures $> 0°$ C ($> 273.15$ K) per day (Hock, 2003) between December, January, and February, using the hourly near-surface temperature data from ERA5. As SGL areas and volumes display the same evolution throughout the study period, we can refer to correlation values from either maximum lake areas (highlighted columns in Sect. 4.3; Table 2), or maximum lake volumes as they would be similar. The ERA5 data used in this study is available at 0.25 degrees resolution, equivalent to ~31 km spatial resolution on the ground at the given latitude. We acknowledge that such coarse spatial resolution is unable to resolve local climatic variations which may be crucial towards influencing the intensity of melt. However, in the absence of any other fine scale meteorological data, these assimilated and modelled products are deemed valuable.

**3.7 Errors and Uncertainties**

Due to absence of any kind of background or ground-truth data, we assessed different sources of errors for a selection of scenes to empirically quantify an uncertainty range for the generated estimates. Firstly, we manually removed obvious false positives by visually comparing lake masks with the underlying RGB imagery, accounting for roughly 0.3 - 0.5% of classified pixels. Major contributors of false positives were shadows, blue ice, nunataks, and crevassed areas. We then compared the automatically generated lake extents with manually digitized lake boundaries for 34 lakes of different sizes and shapes spread over four randomly sampled areas with homogenous and inhomogeneous (slushy) character, broadly representing the supraglacial lake characteristics over the study area. We found very close agreement between the two methods (see Figure S3) with an $r = 0.96$ and RMSE $= 0.51$. Smaller lakes ($<0.01$ km$^2$) had the largest differences in percentage, but due to their small size their contribution to absolute differences was much smaller. Several small water pockets in slushy areas were not classified as lakes during manual digitization but were classified as lakes in the automated approach. In the case of larger lakes, both



methods produced similar areas. On average, the difference in total lake area between the automated and manual approach was
       ~ 0.5 – 0.7 % for each of the four sampled areas. Considering these error sources and ranges, we assign a relative uncertainty
       of 1% to the lake area estimates in our study, similar to Stokes et al. (2019) and Arthur et al. (2020b). Given the strong
       relationship between lake areas and lake volumes (Liang et al., 2012; Trusel et al., 2012), we apply this uncertainty range to
       volume estimates as well.


       Periods with cloud cover and limited satellite overpasses make it difficult to find a series of scenes representing the full melt
       season evolution. In this work, several seasons over the study areas had incomplete time series of data, making multi-seasonal
       assessments difficult. Therefore, we used only peak lake areas and volumes from the melt season that were captured. Judging
       from the climate reanalysis data of near surface air temperatures (Figure 2a), the temperature finally peaks around mid-January,
and the actual peak in lake areas and volumes is captured between 16 – 31 January (Figure 2b). To account for uncertainties
       related to the timing of available data with respect to the actual peak of lake area and volume, we have conceptualized the melt
       season evolution into different stages (Table 1) based on the observed seasonal cycles of lake area which approximates a
       Gaussian distribution with symmetric seasonal growth and decay (Figure 2b). Stage 1 represents the starting and ending phases
       of meltwater ponding, whereas stage 5 represents the period with peak area and volume of SGLs. For each stage, we quantified
the normalized lake area (or volume) using data samples with full seasonal coverage (Table 1). We then outlined 5 different
       scenarios for data availability, ranging from only having data from the initiation and freeze-up periods (stage 1, worst-case

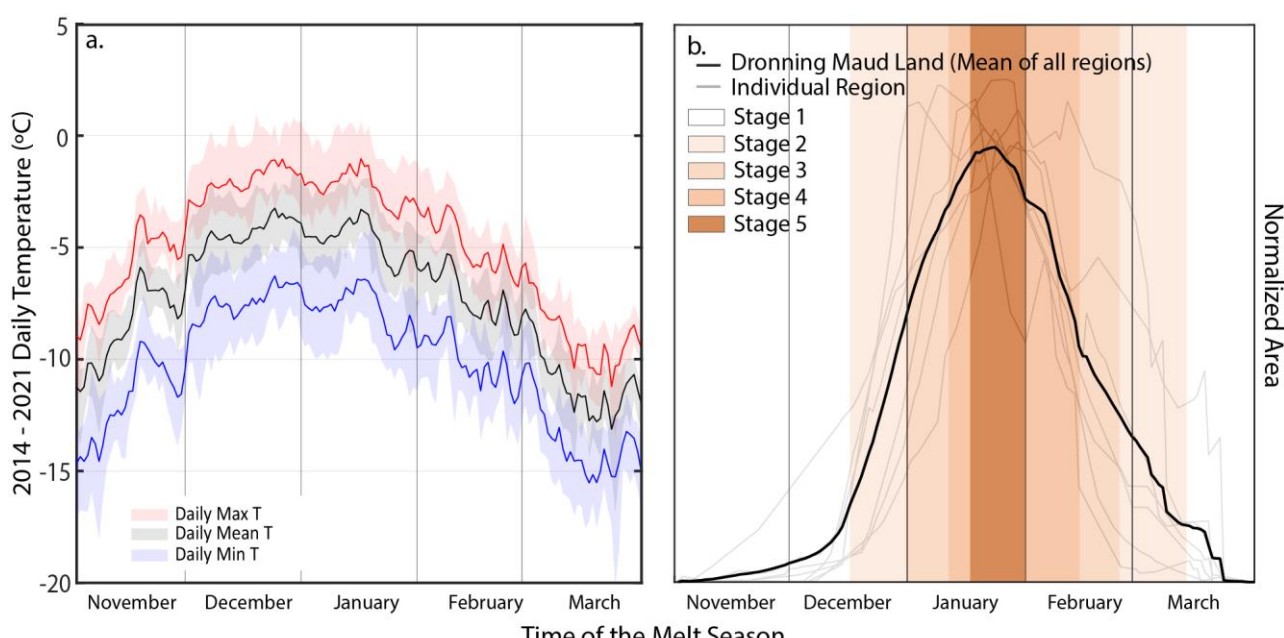

**Figure 2: Daily Near Surface Temperature trends and lake development patterns. a) NDJFM Average Daily Maximum, Mean and Minimum temperatures for Dronning Maud Land; b) NDJFM average normalized lake area plotted against time for each region (grey lines) and overall for Dronning Maud Land (black line). The colours in the background correspond to the stages defined in Table 1.**





scenario) to having data from the period with expected peak area/volume (stage 5, best-case scenario). Based on the normalized lake area/volume for the associated stages, we assigned relative uncertainties for estimated peak area (or volume) for each scenario, ranging from a worst case of 95% (only stage 1 data) to a best case of 5% (including stage 5 data). As an example,

if the latest image we had was from Stage 4, then we assume that 65% of the peak area (or volume) is captured by the scene, and 30-35% of meltwater information remains lost due to absence of a scene from Stage 5 – therefore assuming an uncertainty of 35%. This is irrespective of the number of scenes from previous stages, in this case Stage 1, 2 or 3.

**Table 1. Typical stages and periods for meltwater ponding, advection and freeze-up. Percentage uncertainty for the seasonal peak of lake area and volume is estimated for 5 different scenarios (right columns) based on the availability of scenes from different stages of the melt season. Stage-wise scene availability is defined by 1 = Scene available; 0 = Scene not available. X = Not important as scene from a higher stage is available.**

| Ponding/advection Phases | Melt Stage | Date Range | | Lake Area / Volume Covered % | Accumulated meltwater in particular stage % | Scenarios (left to right) for stage-wise scene availability (top to bottom) | | | | |
|---|---|---|---|---|---|---|---|---|---|---|
| Ponding Initiation Phase | Stage 1 | 01-Nov | 15-Dec | 5% | 5% | 1 | X | X | X | X |
| Increased Ponding & minimal advection Phase | Stage 2 | 16-Dec | 31-Dec | 20% | 15% | 0 | 1 | X | X | X |
| Continued Ponding & enhanced advection phase | Stage 3 | 01-Jan | 10-Jan | 40% | 20% | 0 | 0 | 1 | X | X |
| | Stage 4 | 11-Jan | 15-Jan | 65% | 25% | 0 | 0 | 0 | 1 | X |
| | Stage 5 | 16-Jan | 31-Jan | 95-100% | 35% | 0 | 0 | 0 | 0 | 1 |
| Surface Freezing Phase | Stage 4 | 01-Feb | 15-Feb | 65% | | 0 | 0 | 0 | 1 | X |
| | Stage 3 | 16-Feb | 25-Feb | 40% | | 0 | 0 | 1 | X | X |
| | Stage 2 | 26-Feb | 15-Mar | 20% | | 0 | 1 | X | X | X |
| | Stage 1 | 16-Mar | 31-Mar | 5% | | 1 | X | X | X | X |
| Uncertainty range based on scene availability → | | | | | | 95% | 80% | 60% | 35% | 5% |

# 4 Results

## 4.1 Spatial distribution of SGLs

On Dronning Maud Land, SGLs were identified over five ice shelf regions (Figure 3) - Riiser Larsen, Fimbulisen, Nivlisen, Muninisen and Roi Baudouin. Among these regions Roi Baudouin had SGLs spread over three parts on the ice shelf and we denote them as Roi Baudouin West, Roi Baudouin Center, and Roi Baudouin East.



The SGLs are primarily distributed around the grounding zones, extending laterally over the ice shelves (Figure 3). Production of meltwater occurs mainly above the grounding line, concentrated around the exposed blue ice areas, thereafter meltwater flows downslope following the local topography (e.g., see Figure 4). About ~10-15% of the total SGL area lies on the grounded part of the ice sheet, while the rest extends over the ice shelves. The SGLs are visible as circular ponds and lakes of different sizes, extending up to 30 km$^2$ (even larger when coalesced), and as linear streams with channelized flow of meltwater. Visually,

lake densities are highest on Roi Baudouin and lowest on Muninisen. However, other ice shelves, viz. Nivlisen and Riiser Larsen, also have relatively high densities of SGLs, particularly during years with high melting. Lake depths were generally similar across all the regions, with an overall seasonal mean depth (calculated over the entire melt seasons between 2014 and 2021) of 1.31 m. Among the observed regions, Nivlisen had the highest overall mean depth at 1.94 m. The deepest water body was observed at Roi Baudouin East with a depth of 3.46 m, followed by a 3.41 m deep feature at Riiser Larsen. Both of these

observations are from the peak melting periods, i.e., mid-January. Considering the altitudinal distribution of lakes, the highest

**Figure 3: Spatial distribution and recurrence of SGLs on the 5 ice shelf regions in Dronning Maud Land. The blue colour scale indicates the number of times a particular pixel was classified as water between 2014 and 2021. 1 indicates water only in one of the 7 seasons of observation, whereas 7 indicates water in every melt season between 2014 and 2021.**





number of lakes were present at elevations below 100 m, while the highest lake observed was at 409m above sea level in the Roi Baudouin East region. However, this distribution varied between years with high melt and low melt. During low-melt years, advection of meltwater was limited up to only a few kilometers beyond the grounding zone, presumably due to the low supply of meltwater. Transportation of meltwater occurs primarily though linear and elongated channels or streams that usually

originate near and above the grounding zone and end up in various circular lakes and ponds along the way downstream over the iceshelf. Analysis of lake recurrence (Figure 3) shows that SGLs form a system of channels and usually flow through the same channels year after year. Riiser Larsen, Nivlisen and Roi Baudouin ice shelves have well-established networks of meltwater transportation probably due to their extensive melt regimes from the past, and therefore more water is transported across the ice shelves resulting in lateral expansion of SGLs during the summer season, whereas in case of Fimbulisen and

Muninisen surface transportation of meltwater is minimal, likely due to percolation into the unsaturated firn pack that surrounds the meltwater ponds.

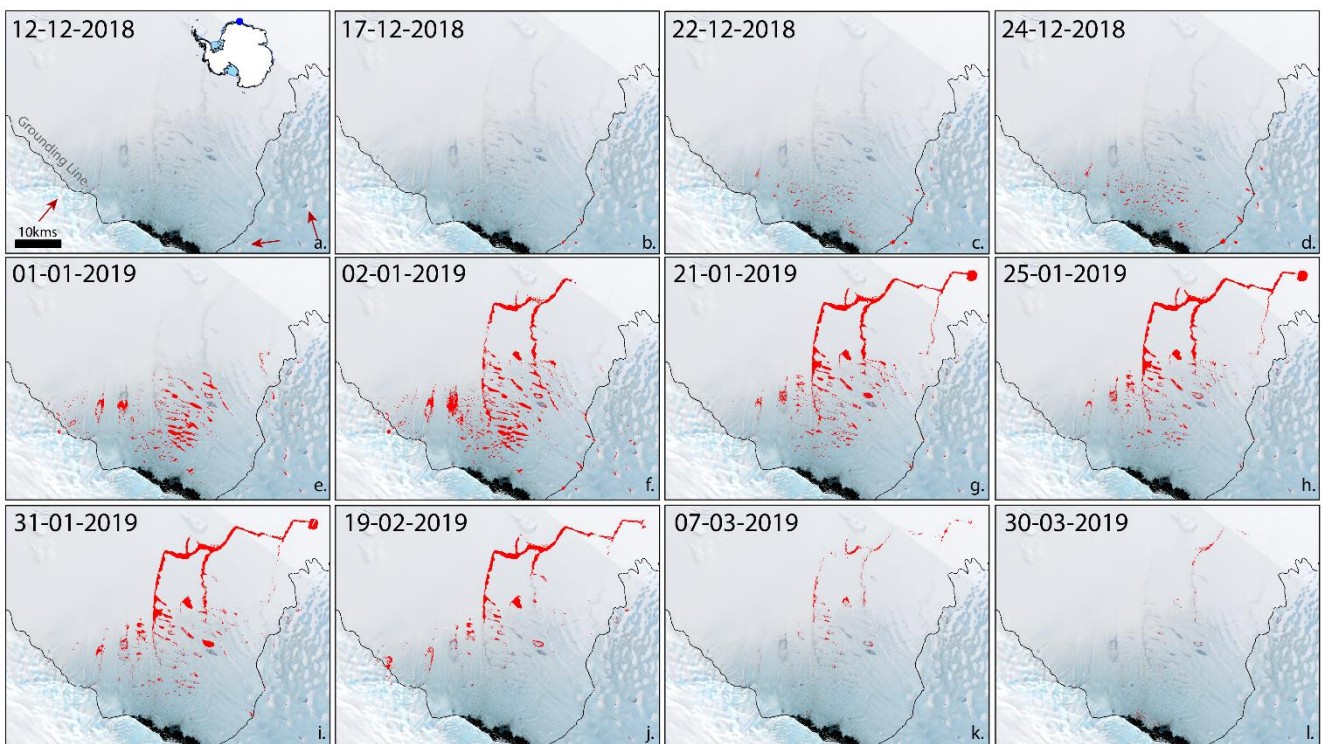

**Figure 4: Intra-Seasonal development of meltwater lakes over Nivlisen (NV) ice shelf in DML during a high melt year (2018 – 2019) with red colour representing the extent of lakes on a specified date. The background image used for representation is a Landsat 8 RGB Composite from 03 November 2017 (Source: USGS).**



## 4.2 Temporal evolution of SGLs

Substantial temporal changes in the SGLs were observed on an intra-seasonal (within a particular melt season) and interannual scale (multiple melt seasons from different years). While the intra-seasonal evolution covers the growth and decay of SGLs in a particular melt season, the interannual evolution explains how the SGLs have varied over longer time periods.

### 4.2.1 Seasonal evolution

Meltwater ponding is first observed between mid-November and early-December across all regions in the study. However, as
the magnitude of melt is low during this period (Johnson et al., 2021; Trusel et al., 2012), ponding is minimal (Figure 4a). As melting intensifies, meltwater accumulation increases from mid-December onwards, mainly in small depressions over the areas close to and above the grounding zone (Figure 4b-d) where meltwater production is more efficient due to the exposed blue ice having lower albedo. As these depressions are small in scale, they soon start to overflow, advecting meltwater through streams over and across the grounding zones to the flatter ice shelves. A significant rise in surface ponding is recorded between
early & mid-January (Figure 4e-i), and shortly after we capture the peaks of meltwater ponding, i.e., when areas and volumes of SGLs are at the highest (mid-January to end-January). During this period, significant advection of meltwater occurs across

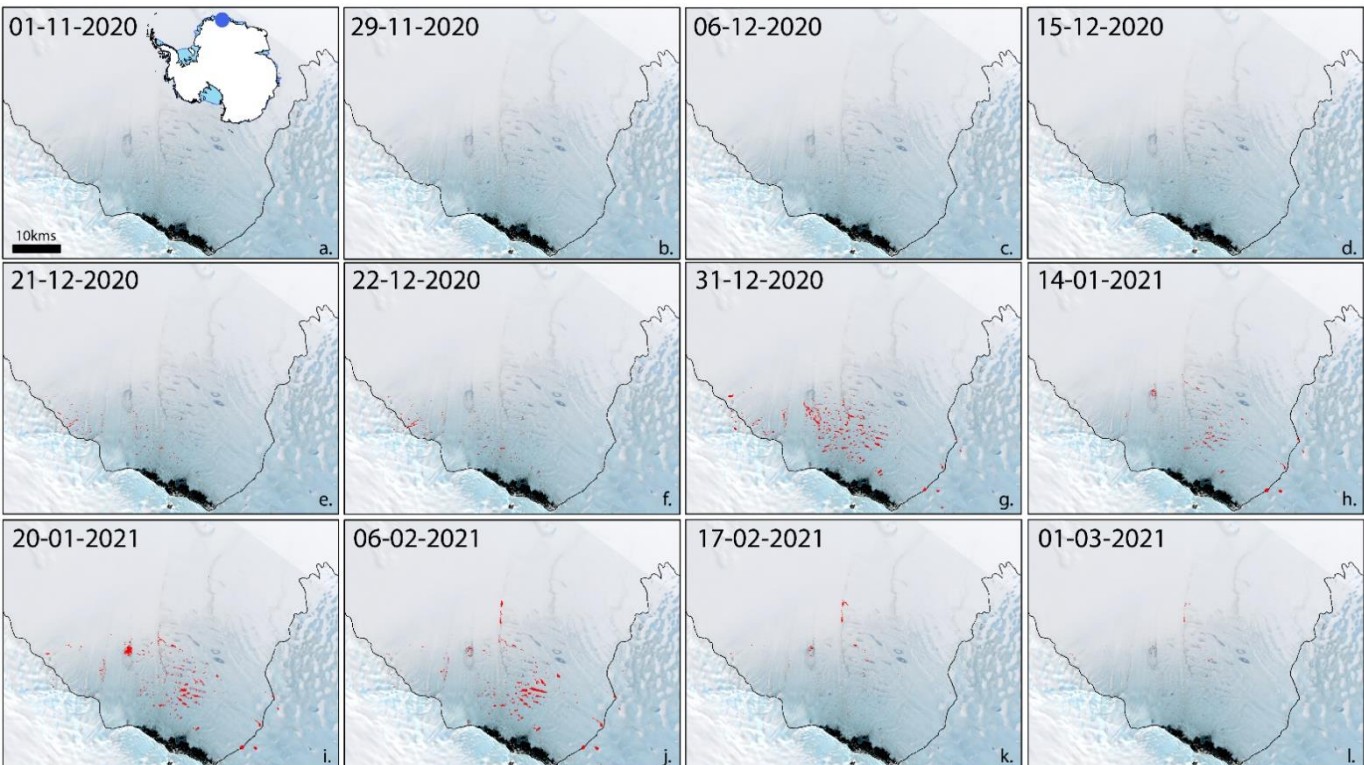

**Figure 5: Intra-Seasonal development of meltwater lakes over Nivlisen Area in Dronning Maud Land during a low melt year (2020 – 2021) with red colour representing the extent of lakes on a specified date. The background image used for representation is a Landsat 8 RGB Composite from 03 November 2017 (Source: USGS).**

the ice shelf. Thereafter, as air temperatures drop (Figure 2), SGLs begin to freeze (Figure 4j-l) with thin layers of ice appearing on lake surfaces, continuing until end of March when almost no surface meltwater is present. The evolution of meltwater ponding during high melt years and low melt years is very different. Due to lower meltwater availability, ponding and advection of water is minimal during low melt years (e.g., 2020-2021 melt season over Nivlisen; Figure 5), whereas on high melt years (e.g., 2018-2019 melt season over Nivlisen; Figure 4) ponding is rapid and widespread over the area. Another set of examples from Riiser Larsen are presented in the supplement (Figure S4 and Figure S5), affirming the same trends of intra-seasonal evolution between high and low melt years in the region.

### 4.2.2 Interannual evolution

Large variability in surface ponding is seen from year to year over all areas where SGLs were identified and assessed. The variations in SGL areas and volumes between 2014 and 2021 and their deviation from the average estimates of the study period are presented in Figure 6. We see that 2015-2016 and 2020-2021 are years with insignificant surface ponding (Figure 6; highlighted with green), whereas 2016-2017 and 2019-2020 are years where large ponding is observed (Figure 6; highlighted with red). The exception of Roi Baudouin East in 2019-2020 is attributed to uncertainty from unavailable cloud free images. Nevertheless, this magnitude of surface ponding, high or low, is relatively similar across all regions in Dronning Maud Land, indicating the existence of at least one common factor controlling melting throughout the region.

Annual anomalies of SGL area (or volume) with respect to the mean of 2014-2021 helps in understanding interannual variations between regions (Figure 6b). The largest SGL anomaly was observed over Muninisen in the 2016-2017 melt-season with an increase of ~200% in SGL area compared to the mean, whereas those over Fimbulisen and Riiser Larsen have the largest positive anomalies (~150%) in 2018-2019. Nivlisen, on the other hand, had the largest SGL anomaly in 2015-2016 with 90% lower area and volume. Interestingly, lakes from the three parts of Roi Baudouin ice shelf behaved differently in different years. While lakes in Roi Baudouin Center had the largest anomaly in the 2019-2020 melt season (~100% higher area and volume), Roi Baudouin West and Roi Baudouin East had the largest anomaly in the 2016-2017 melt season (about 120% higher in both cases). This mixed pattern in nearby regions suggests that local processes also influence melting and ponding in the region.





**Figure 6: Temporal variation of SGL Area and Volume over 5 ice shelves of Dronning Maud Land. a) Quantitative variations in area and volume over 7 melt seasons between 2014 and 2021 with error bars in blue. b) Anomaly with respect to 2014-2021 averages. The red and green highlights represent the high and low melt seasons respectively.**



### 4.3 Relationship between SGL evolution and climate

The correlation between near-surface temperature and SGL extents (Table 2) is inconsistent between regions in Dronning Maud Land. There is strong positive correlation over some of the regions (Riiser Larsen, Nivlisen and Roi Baudouin Center), whereas over other regions the correlation is not as strong (Muninisen, Roi Baudouin West and Roi Baudouin East). This regional inconsistency in correlation is similar for the relationship with PDD too. We see that warmer years, either with high DJF temperature or high PDD, don't necessarily correspond to years with higher melting and ponding extents. The 2018-2019

melt year is one of the years with relatively high DJF temperature and PDD, but with average melting and ponding on the surface. On the contrary, the low-melt year 2020-2021 has relatively low ranges of temperature and PDD, concurring with lower SGL extents. This is in contrast to the 2015-2016 low-melt year which was comparatively warmer but had similar low extents of SGLs. Fimbulisen stands out as the region with warmest near-surface DJF temperature and highest PDD (See Figure 7b), yet it has the lowest extents of SGLs in all regions. In comparison, the adjacent ice shelf Nivlisen has considerably higher

SGL density, but similar near-surface temperature range and lower PDD (Figure 7a & 7g, respectively). The reasons for these large variations in surface ponding over different regions (over 2 orders of magnitude) remain unexplained.

For the melt years 2016-2017 and 2019-2020 (highlighted red, Figure 7) all the areas in Dronning Maud Land experienced above average ponding of meltwater, whereas in 2015-2016 & 2020-2021 (highlighted green, Fig. 8) the ponding was minimal.

This relative consistency in melt extents for different areas indicates that there are common climatic factors that control melting and ponding in Dronning Maud Land at large.

**Table 2: Correlation ($r$) between SGL extents (max area and max volume), DJF mean temperature (ºC) and Positive Degree Days (PDD) in DJF on different ices helves in Dronning Maud Land.**

| Area | Correlation ($r$) between | | | |
|---|---|---|---|---|
| | Max Area & DJF Mean Temperature | Max Volume & DJF Mean Temperature | Max Area & PDD in DJF | Max Volume & PDD in DJF |
| **Riiser Larsen** | 0.75 | 0.71 | *-0.08* | *-0.10* |
| **Fimbulisen** | 0.61 | 0.67 | 0.87 | 0.90 |
| **Nivlisen** | 0.72 | 0.74 | 0.48 | 0.49 |
| **Muninisen** | 0.61 | 0.61 | 0.76 | 0.78 |
| **Roi Baudouin West** | 0.53 | 0.47 | 0.54 | 0.46 |
| **Roi Baudouin Center** | 0.70 | 0.76 | 0.87 | 0.91 |
| **Roi Baudouin East** | 0.34 | 0.39 | 0.66 | 0.72 |
| **Dronning Maud Land** | *-0.07* | *-0.02* | *-0.13* | *-0.12* |


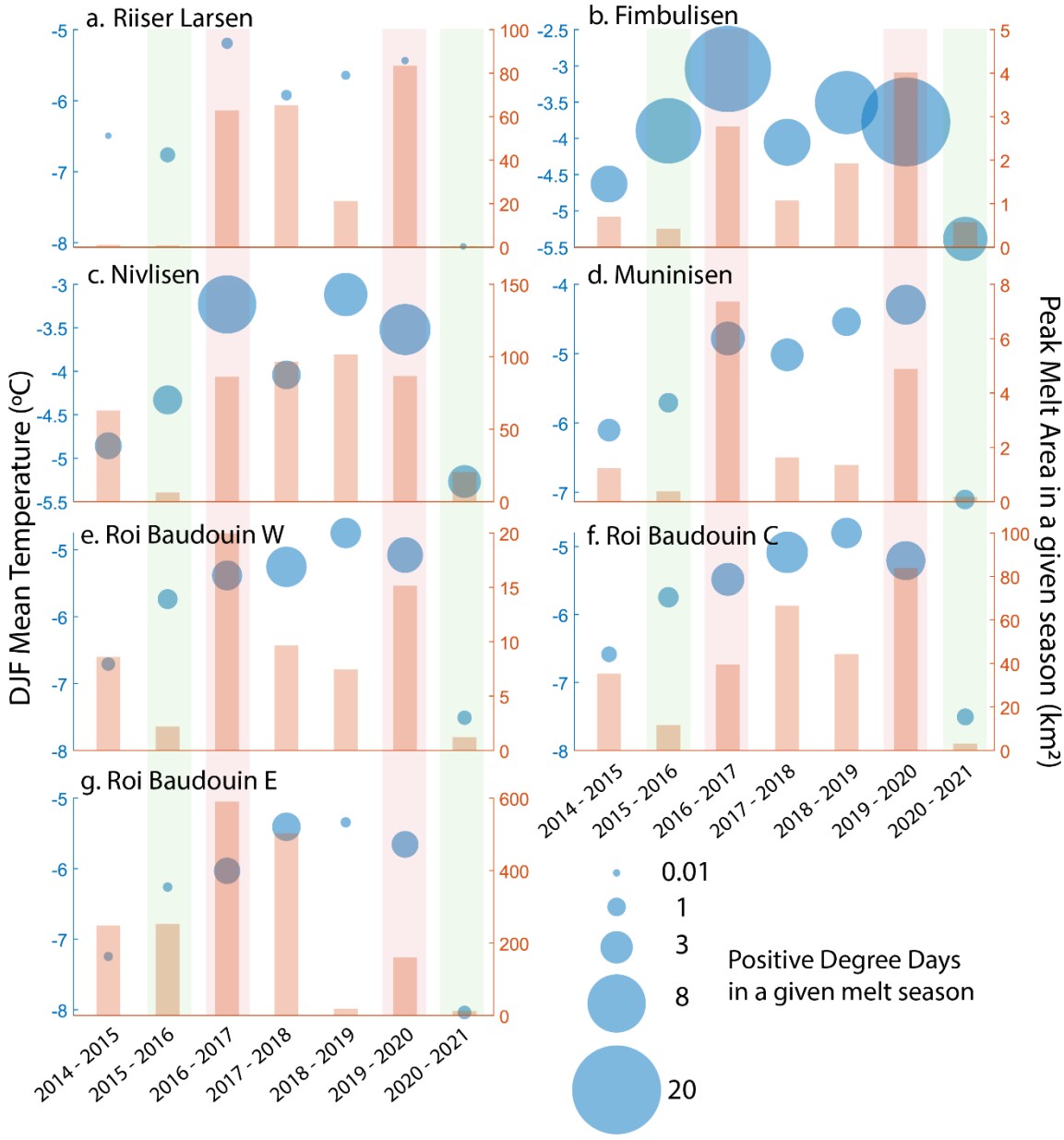

**Figure 7: DJF mean temperatures (left axis), PDD (circles) and Peak SGL Area (right axis) over the studied ice shelves of Dronning Maud Land. The bars represent the peak area covered by melt lakes during a particular melt season; the location of the circles corresponds to the average DJF Mean Temperature and the size of the circles refers to PDD. The red and green background highlights four years with anomalously high and low melt extents, respectively.**





## 5 Discussion

### 5.1. Distribution of SGLs

SGLs are found in five main ice shelf regions in Dronning Maud Land, but with highly variable distribution and extents. For instance, Riiser Larsen, Nivlisen and Roi Baudouin regions have lakes distributed over the grounded ice sheet as well as on the floating ice shelves, whereas lakes on Fimbulisen and Muninisen are only present over the grounded ice sheet. This may be attributed to local topography and climate, or to a more extensive firn pack where meltwater would percolate and refreeze.

The number and size of melt lakes and ponds over Fimbulisen and Muninisen are indicative of lower melting in these regions, which is consistent with independent studies of surface melt extent and duration from satellite scatterometry (Trusel et al., 2012) and passive microwave data (Picard et al., 2007). However, the climatic factors that explain these regional differences are yet to be identified.

SGLs occur intermittently during the austral summer months when atmospheric conditions are favorable for melting and when there is no pore space in the snow or firn to absorb it. The rise of near-surface temperature from mid-November marks the onset of the melt season that lasts until end of January, consistent with the melt durations derived for Dronning Maud Land by Johnson et al. (2021) and Trusel et al. (2012). By the end of the melt period, the surface meltwater has drained or begins to freeze at the surface to form relict lakes. Over the entire study area, SGLs are found to cluster around the grounding zone, the

zone of transition from grounded ice sheet to floating ice shelf, often characterized by a break in slope from the steeper inland to the flatter ice shelves. Concentration of lakes in this area might be attributed to persistent katabatic winds (Lenaerts et al., 2016) that cause localized warming as the downslope air-flow interacts with the surface. Most of the SGLs above the grounding zone are smaller in size as they form in small topographic depressions (see Fig. S6) and quickly overflow to fill up other depressions down the slope. Due to the overall slope, ponding area is limited and therefore large lakes are non-existent on the

grounded ice sheet. However, it is believed that most of the meltwater production occurs in this region (Lenaerts et al., 2014) where there are strong katabatic winds and extensive blue ice areas. In this zone, two processes contribute to generation of meltwater: first, the adiabatic warming of descending winds that locally enhances the surface temperature causing snow, firn, and ice to melt and second, the albedo-lowering feedback mechanism of exposed blue ice areas that enhances melting of surrounding ice. The produced meltwater is laterally transported across the grounding zone and over the ice shelves by surface

streams and channels. Existence of subsurface meltwater channels cannot be ruled out as observed and described by Winther et al. (1996) from their field surveys in the 1990s. They describe the existence of a 'solid-state-greenhouse effect' (Brandt and Warren, 1993; Schlatter, 1972) resulting from penetration of shortwave solar radiation to underlying snow/firn layers while longwave radiative cooling is restricted to the surface, thereby explaining the possibility of active subsurface meltwater network. However, this cannot be investigated with our satellite data.




Meltwater that gets transported to the flatter ice shelves tends to pond over larger depressions forming lakes ranging from 5 km$^2$ to 30 km$^2$ or more when coalesced together as can be seen on Riiser Larsen, Nivlisen and Roi Baudouin ice shelves. Fimbulisen and Muninisen seem to have too small volumes of meltwater or too much pore space in the firn for large advection to occur over the ice shelves. However, repeated formation of lakes over the same locations implies the existence of a locally

saturated firn layer, refrozen ice lenses or superimposed ice (Hubbard et al., 2016). The absence of meltwater lakes near the front of the ice shelves in Dronning Maud Land is explained by limited meltwater production compared to snow accumulation, which builds a firn pack that is able to absorb and refreeze all meltwater, while keeping the albedo high and melting low compared to the blue ice areas.

The surface depressions that form lakes on the grounded ice sheet are a representation of subsurface undulations and are mostly unaffected by the movement of ice, whereas lakes over the floating ice shelf move along with ice-flow, thereby being displaced in space despite occurring in the same relative location. Due to saturation of the firn from past melting and low accumulation around the grounding zone, ponding gets initiated early in the melt season, and the network of lakes and streams remains intact over years, making the development of the meltwater network faster in successive years when compared to areas where a new

network would be established in an event of excessively high melt. This recurrence pattern is similar to what has been observed on other ice shelves in Antarctica (Banwell et al., 2014; Luckman et al., 2014; Reynolds, 1981; Arthur et al., 2020a; Tuckett et al., 2021).

### 5.2. Evolution and Variability of SGLs

We observe a characteristic pattern of seasonal evolution, from limited ponding in the beginning of the melt season to rapid

increase between late-December and early-January to reaching the maximum extents around mid-January (see Figure 2b). From early-February onwards, surface lakes begin to freeze partially or completely (further discussed in Sect. 5.3). This consistent seasonal pattern is similar to that observed over SGLs of Amery Ice Shelf (Tuckett et al., 2021). On the other hand, variability in SGL extents is high both on seasonal and interannual scales, consistent with what is reported by Arthur et al. (2022) for other parts of the East Antarctic Ice Sheet. Some years stand out with anomalously high or low extents of lakes

across Dronning Maud Land (see Fig. 3). During the high melt years (2016-2017 & 2019-2020), melting is further intensified by the feedback mechanisms contributing to larger differences in the extents of ponding, whereas during low melt years (2015-2016 and 2020-2021) strong feedback mechanisms are not triggered. The occurrences of high and low melt years appear random within our limited study period, with no particular trend being observed during the study period - 2014-2021. Other studies that looked at longer time periods (e.g. since 2000s) have also reported large variations and no clear trend for SGLs

around different ice shelves areas of Antarctica (Arthur et al., 2020a, 2022; Tuckett et al., 2021; Langley et al., 2016; Moussavi et al., 2016). During high melt years, lateral transfer of meltwater across the grounding zone over the ice shelf is high, whereas in low melt years, meltwater transfer through surface streams is minimal. During these years, lakes generally are widespread





near the grounding zones, and very few lakes are seen over the shelves (e.g., Figure 5 and S5). The high melt year 2016-2017 witnessed wide-spread ponding over the surface immediately after a low melt year (2015-2016) with insignificant ponding,

and vice-versa in case of 2019-2020 and 2020-2021, implying that the delay in ponding due to meltwater filling of pore spaces in preceding years' snow and firn is limited in our study regions, presumably due to the extensive blue ice areas that stay snow-free most of the year. However, given the short time series, this cannot be statistically established and more such events are required to gain confidence.

### 5.3 Lake Freeze-up & Disappearance

Given the dynamicity of meltwater and its sensitivity to atmospheric conditions, SGLs are present on the surface only for a particular period when the conditions are most conducive, i.e., the austral summer. The lakes continuously evolve as the melt season initiates and eventually disappear at the end of the season. The disappearance of lakes can happen in different ways (Arthur et al., 2020a; Lenaerts et al., 2016; Dell et al., 2020; Williamson et al., 2018): draining into firn/snow areas or lakes

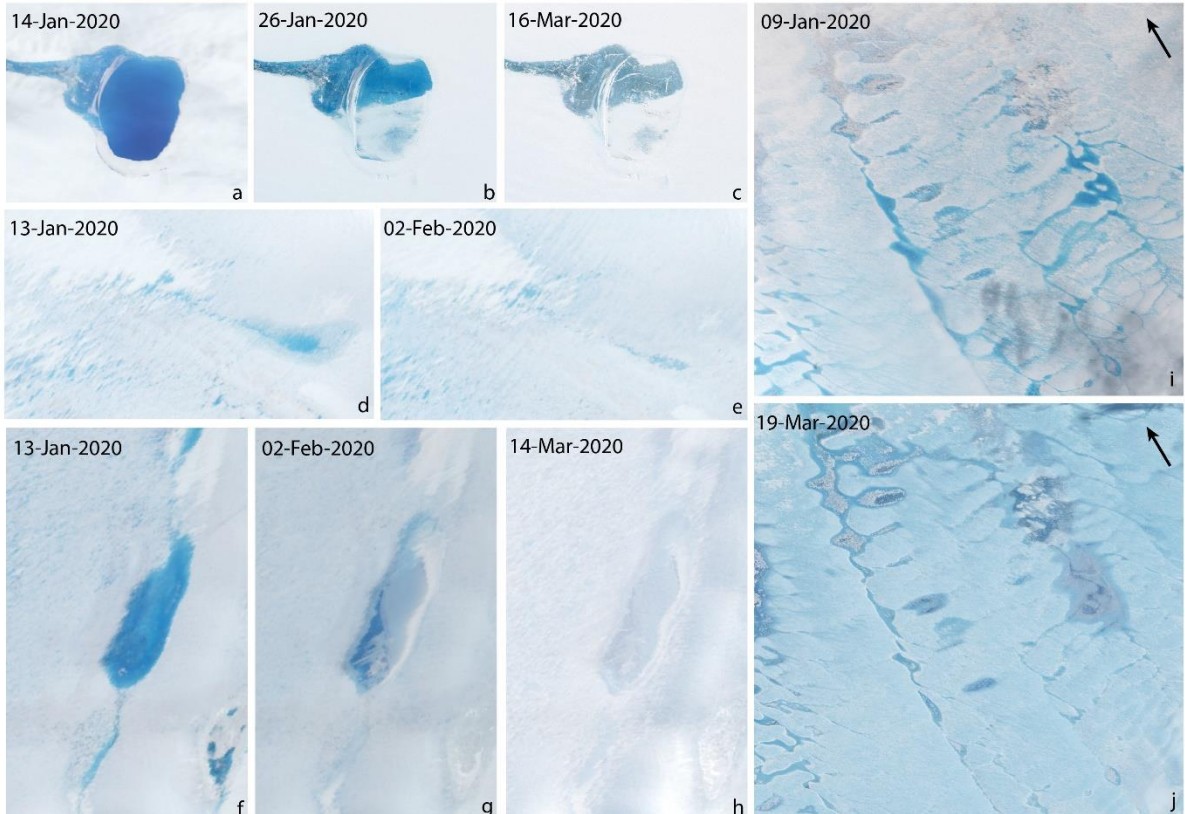

**Figure 8: Freeze-up and disappearance of SGLs seen over different ice shelves in Dronning Maud Land. a-c: Surficial freezing of deep lakes. The surface ice layer potentially insulates deeper water; d-e: draining of lakes into surrounding firn/snow pack; f-h: complete freezing of shallow lakes; i-j: draining of lakes into lakes downstream/downslope through supraglacial channels. Arrows indicate the flow direction or downstream direction. Images are clipped from Landsat-8 (a, j; Source: USGS) and Sentinel-2 (b, c, d, e, f, g, h, I; Source: Copernicus Open Access Hub) scenes from dates mentioned on respective panels.**





downstream, subglacial drainage through crevasses/fractures, directly draining into the ocean through surface streams, or for
pooled water; partial surficial freezing and subsequent insulation of deeper meltwater and complete freezing of the lakes. Close
observation of satellite images from the study areas reveals that drainage into surrounding snow/firn pack and/or lakes and
streams downstream is among the first events of drainage during the melt season (Figure 8d-e and Figure 8i-j). This type of
draining occurs throughout the melt season, usually in lakes that lie close to the grounding zones. As air temperatures begin to
drop, generally after mid-January (See Figure 2a), the upper water layer of most of the SGLs begins to freeze forming a thin
layer of ice on the surface, gradually continuing to freeze thicker. The extent to which the SGLs freeze to the bottom is unclear.
Based on visual interpretation from Figure 8a-c and Figure 8f-h, we think that the deeper lakes freeze partially insulating the
water at the bottom (which might freeze later during winter), whereas the shallower lakes freeze completely within a few
months. As snowfall occurs, many of these lakes get buried under a fresh snow layer, causing further insulation of underlying
meltwater and precluding further observations from space. Detailed field or modelling studies would be needed to further
determine conditions of insulation and freeze-up. Although the extent of aquifers in firn or ice is expected to be small in the
region, there have been observations of some englacial meltwater networks at progressively increasing depths towards the
calving front from the grounding line along the western Roi Baudouin area (Lenaerts et al., 2016). No evidence of subglacial
drainage of lakes though crevasses/cracks or englacial conduits has been observed in Dronning Maud Land, but considering
recent observations elsewhere in East Antarctica (e.g., over Amery Ice Shelf (Spergel et al., 2021)), it could be plausible.
Direct surface drainage into the ocean is less likely to happen as all identified SGL areas are more than several tens of
kilometers from the ice shelf fronts.

**5.4. Environmental and climatic controls**

Near-surface temperatures are expected to be closely linked with the production of surface-meltwater on ice shelves. However,
from the coarsely resolved temperature reanalysis data, we see no clear dependency on temperature with no correlation ($r =$
~0) for Dronning Maud Land as a whole. On regional scales we see a higher degree of correlation ($r =$ ~0.7) for Riiser Larsen,
Nivlisen and Roi Baudouin Center, but the large interannual variations in peak SGL area and volume (2 degrees of magnitude)
cannot be easily explained. This lack of correlation may be due to the coarse spatial resolution of ERA5, which is ~31 km. At
this scale, climate models fail to capture localized processes and feedbacks that may have a crucial role in influencing surface
melting and production of meltwater (Arthur et al., 2020a; Lenaerts et al., 2016). For Muninisen, Roi Baudouin West and Roi
Baudouin East we see relatively weaker correlations between temperature and SGL area/volume, similar to that reported by
Arthur et al. (2020) over Shackleton ice shelf. This is in contrast to Langhovde glacier further east of Roi Baudouin where a
strong temperature-SGL correlation was found (Langley et al., 2016). But this study used near-surface temperature measured
by an automatic weather station, only a few kilometers away from the glacial lakes, being able to capture localized temperature
thereby contributing to higher confidence in the temperature – SGL relationship. This underscores the need for more field
observations and high-resolution climate modeling over Dronning Maud Land and other parts of Antarctica (Arthur et al.,
2020b; Stokes et al., 2019).



There are many different factors that can influence surface melting and ponding of meltwater. For example, climatic factors such as katabatic winds can cause adiabatic warming as they descend from higher elevations, precipitation events can cause
release of latent heat, and presence of clouds can intensify local warming by increasing emissivity and directing longwave radiations towards the ground (Kittel et al., 2022). Additionally, environmental (physical) factors such as blue ice can trigger a feedback mechanism that enhances melting due to the low albedo, differences in firn air content can influence the availability of meltwater for ponding, and surface topography can influence the flow and spread of meltwater. Fohn winds have also been observed to promote melting and ponding over Larsen C ice shelf in the Antarctic Peninsula (Luckman et al., 2014), however
these types of winds do not form in Dronning Maud Land (Lenaerts et al., 2016). Katabatic winds, which are prevalent in coastal East Antarctica, are also very impactful as they can cause >3K warming (Lenaerts et al., 2016) compared to their surroundings. Nunataks have an added albedo-lowering-feedback due to their darker appearance, and can have highly localized effects on melting (within 10s of km; Kingslake et al., 2017; Stokes et al., 2019; Winther et al., 1996). However, the SGL systems in Dronning Maud Land are 80-100 km away from the nearest rock outcrops, except near the grounding zone of
Nivlisen where Schirmacher Oasis is an exposed rock area that is roughly 25 km long and a few km wide. Several small pockets of melt ponds can be seen around this oasis, possibly related to the low-albedo feedback. More detailed studies are needed to confidently determine the key factors controlling meltwater production and ponding in Dronning Maud Land.

**5.5. Implications on the stability of ice-shelves**

Previous studies have shown the effect of surface meltwater ponding on firn air depletion, hydrofracture and collapse.
Meltwater induced firn air depletion is considered a precursor for triggering hydrofracture (Kuipers Munneke et al., 2014) of ice shelves, potentially leading to a loss of buttressing and consequent speed-up and enhanced mass loss from the ice sheet. Given the projections of future global warming, an increase in surface melting and ponding is expected over Antarctic ice shelves (Bell et al., 2018; Trusel et al., 2015; Kingslake et al., 2017). However, given the large variability ice shelf characteristics and observed SGL systems, it is important to assess the vulnerability and risks of fracture for each of these ice
shelves individually based on factors such as surface meltwater accumulation, firn air depletion, buttressing potential and thickness of the ice shelves. Lai et al. (2020) assessed the vulnerability of Antarctic ice shelves to hydrofracture using a deep learning framework with liner elastic fracture mechanics (LEFM; van der Veen, 1998), and reported that 60 ± 10% of the Antarctic ice shelves with high buttressing potential are currently vulnerable. Such shelves are also present on Dronning Maud Land, East Antarctica (Fig. 4 from Lai et al. (2020)) implying the importance to better understand the evolution of these
meltwater systems over vulnerable buttressing regions for improved prediction of ice-sheet change in the future .





## 6 Conclusions

We present the first multi-year (2014-2021) intra-seasonal and interannual assessment of development and evolution of SGLs over ice shelves in Dronning Maud Land by exploiting the synergies between Landsat 8 and Sentinel-2 products. Nearly 2500 satellite scenes were examined to create 581 SGL products, each representing a particular ice shelf area on a specific date.

Significant SGL networks were identified over five regions in Dronning Maud Land – namely Riiser Larsen, Fimbulisen, Nivlisen, Muninisen and Roi Baudouin. Amongst these, Roi Baudouin had the highest concentration of meltwater ponding on the surface, spread over three regions of the ice shelf – the west, center and east. Riiser Larsen and Nivlisen also had large networks of lakes and streams in years of high melting, whereas Fimbulisen and Muninisen had relatively fewer and smaller meltwater ponds primarily concentrated above the grounding zone. The 7-year observational record reveals large variability

in lake extents from year to year, whereas patterns of seasonal evolution are more consistent. Ponding is negligible at the beginning of the melt season, mostly clustered around the grounding zone in small topographic depressions, which upon intensification of melt is advected over the ice shelves by surface streams. The lakes reach their peak extents between mid to late-January after which they start to freeze, either partially or completely based on their depths. We did not find any evidence of englacial drainage into crevasses/fractures, nor any indications of direct drainage to the oceans over any of the study areas.

Upon assessment of SGL anomalies with respect to 2014-2021 averages, the 2016-2017 and 2019-2020 melt years stood out with widespread accumulation of meltwater on the surface, whereas 2015-2016 and 2020-2021 were years with low melting and ponding, indicating some large-scale control by climatic factors Although correlation with positive degree days and DJF mean temperatures in the summer season (December-February) was strong in some regions, it did not explain the large differences in magnitude of SGLs in different regions having similar near surface conditions. For example, surface ponding

varies strongly between different parts of Roi Baudouin ice shelf. This could be due to the coarse resolution of temperature data that we used (ERA5, ~31kms), failing to resolve localized conditions, but it could also mean that other climatic factors have a stronger control on melting and ponding in the region, viz. katabatic winds, exposed blue ice areas and associated albedo feedback, or precipitation. To further examine that, there is a need for glacio-climatological field observations from the lake areas in combination with higher resolution modeling of governing processes.

*Code and data availability.* The MATLAB scripts for mapping lake extents and estimating lake depths using L8 and S2 products are published via GitHub (accessible from https://github.com/anirudhavm/SGL-Mapping-L8-S2). All SGL products generated from the satellite imagery are published at the Norwegian Polar Data Centre (https://data.npolar.no/dataset/31aae21f-7465-4b36-ab95-433d7657d5f2). Raw imagery of L8 and S2 is available from USGS Earth Explorer Repository (https://earthexplorer.usgs.gov/) and Copernicus Open Access Hub Portal

(https://scihub.copernicus.eu/). ERA5 Climate Reanalysis data can be access from the Copernicus Climate Change Service (C3S) Climate Data Store (CDS) Portal (https://cds.climate.copernicus.eu/cdsapp#!/dataset/reanalysis-era5-single-levels; Copernicus Climate Change Service (C3S), 2017).



**Author Contributions.** AM & GM conceptualized the idea and designed this study. AM carried out the analysis under the guidance of GM and TVS. AM prepared the manuscript. GM and TVS reviewed and helped improve the manuscript.

**Competing interests.** The authors declare no conflict of interest.

**Acknowledgements.** This work is a part of PhD fellowship that AM received from the Ministry of Earth Sciences (MoEF), Govt. of India through the National Center for Polar and Ocean Research (NCPOR). The authors acknowledge and appreciate the open data policy of USGS and European Space Agency, from which the Landsat-8 and Sentilel-2 SGL datasets were procured. The authors also acknowledge the Polar Geospatial Center (PGC) for making REMA products freely available.

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
