# Peer review of "Recent Evolution of Supraglacial Lakes on ice shelves in Dronning Maud Land, East Antarctica"

_The Cryosphere, 2023_

## Referee Comment (RC1)

Review of Mahagaonkar et al: Recent Evolution of Supraglacial Lakes on ice shelves in Dronning Maud Land, East Antarctica.

This study investigates the intra-seasonal and inter-annual evolution of supraglacial lakes in Dronning Maud Land, East Antarctica. The authors detect supraglacial lakes in Landsat-8 and Sentinel-2 satellite imagery from 2014-2021 for five ice shelf regions. They compare SGL area and volume with near-surface air temperature and positive degree days from the reanalysis model, ERA5 to climatologically explain SGL variability.

Overall, this manuscript is well-written and the figures are clear and informative. However, I have concerns about this paper's contribution to the field. Generally, it does not provide new information that is not already presented in other studies. Given the more specific comments listed below, I believe that major revisions are required.

**Major comments**

1)      There are many previous studies that present surveys of supraglacial lakes for a specific region (Stokes et al., 2019 does so for all of East Antarctica, including DML). As such, this work feels incomplete without further investigating the climatic controls on supraglacial lake evolution in DML, especially since this is listed as an objective of this study (L65).

Further, the discussion section reads more as a literary review, providing little new information and insight into SGL evolution. Authors mention several times that the climatic factors that explain SGL development and evolution differences are still unknown and should be further investigated (i.e. L336, L359, L447, L471). I believe that this study should be expanded to provide additional insight into these climatic controls, especially for regions where air temperature and positive degree days do not explain SGL variability.  For example, in L310 – what is the common factor controlling melting throughout the region? Similarly, in line 320, what are the local processes that also influence melting and ponding? This study feels incomplete without a further investigation into these climatic controls and local processes that impact meltwater ponding in DML.

2)      The citations (especially in the introduction) are outdated and incomplete. Some specific examples are listed below:
   - L30 – Is there a newer source that could be cited here? i.e. Johnson et al., 2022
   - L34 – Banwell and MacAyeal 2015 is a more appropriate citation for ice shelf flexure and fracture.
   - L34 – Please replace the DeConto and Pollard 2015 citation with: Banwell et al., 2013 and Scambos et al., 2009
   - L35 - Please add a citation for "increased buttressing"
   - L36 – This Kuipers-Munneke 2014 paper is on firn air depletion, not firn aquifers please cite (**https://doi.org/10.1002/2013GL058389**) instead.

- L41 – There are several newer studies that look at the pervasive meltwater ponding on the Antarctic Peninsula (Leeson et al., 2020, Banwell et al., 2021)
- L46-48 - Langley et al., 2016 look at the seasonal evolution of supraglacial lakes on an outlet glacier in DML and should be mentioned here (also in L87-89).
- Lenaerts et al., 2016 should be Lenaerts et al., 2017
- L53 – 60 – many citations are missing in this paragraph including: Leppäranta et al. 2013, Liston et al. 1999, Dunmire et al. 2020
- L58 – Add Moussavi et al. 2020
- L77 – Add Bell et al. 2017
- L84 – Add Lenaerts et al. 2017
- Dunmire et al. 2020 should be cited in several places (L426 for "partial surficial freezing and subsequent insulation of deeper meltwater", L437-440 for DML lake drainage).

3)      Finally, there are several other satellite and in-situ observations that could be utilized to expand this analysis. I believe that using microwave imagery (e.g., from Sentinel-1) would help paint a more complete picture of DML lakes. For example, in L379, can this be investigated with microwave imagery? Additionally, does the seasonal evolution pattern you observe throughout the melt season match backscatter signals from Sentinel-1 indicating surface melting (L399)? Because the seasonal evolution is a large part of this paper, I believe it would be beneficial to compare with Sentinel-1, a satellite with year-round frequent observations. Sentinel-1 observations could be further used to see when/if lakes freeze completely (L430-435).

Further, extensive in-situ meteorological observations are available from Neumeyer station in DML. Why were these observations not utilized?

**Minor comments**

L17 – Please specify which ice shelves showed "no significant meltwater lakes"

L29 – Specify that runoff is not significant *on Antarctica*

L49 – What do you mean by "major" ice shelves? Buttressing capacity? Size?

L59 – What do you mean by "simple configuration"?

L78 – 80 – No need to cite Trusel et al. 2013 twice in this same sentence.

L80 – 82: Please remove the sentence: "The ice shelves of Dronning… East Ragnhild Glaciers" as it is not necessary.

L98 – 100: Please reword this sentence beginning with "We did not use the cloud detection…" as it is confusing.

Why were the 5 ice shelves used in this study chosen over other ice shelves in DML?

Sections 3.2-3.3: It is unclear to me where you followed methods from other papers and where you did not. For example, in lines 117-22, was this still following Moussavi et al 2020 or did you use different thresholds? If the method is the same as work previously published, then the text can be simplified greatly. If not, please explain why you chose not to follow previously established methods. Also, are there any periods for which data is lacking? If so, please explain.

L141 – Why do you use a threshold to exclude shallow lakes? I would think that by excluding shallow lakes you lead to an underestimation of SGLs as shallow lakes are still lakes!

Section 3.4 – It is my understanding that the estimation of lake depth and volume is largely based off previously published methods? If so, this section can be greatly reduced.

Figure S1 – Why does Landsat 8 estimate greater depth for deeper lakes?

Section 3.6 – why do you use ERA5 here?

L219 – How were the manually digitized lake boundaries created? Were they created completely independently from the automatic lake masks?

L228 – I believe that uncertainty with depth calculations should be considered as well in the uncertainty range for lake volume.

L240 – 250 – This section is a bit confusing to me and I think could be reworded for simplicity.

Table 1 – What is the vertical curve on the left-hand side of the table (over the ponding/advection phases column)?

L 258 – Refer to Figure 3 for the different ice shelf regions.

L265 – What do you mean by "viz."? (also L507)

L266 – Specify what are the years with high melting.

L268-269 – Please include error when quantifying lake depths.

Figure 3 – I think it would be nice to also include the blue ice areas in this figure.

Line 280 – More specifically, what is different about the Fimbulisen/Muninisen firn pack? Less blue ice? More firn air content?

Figures 4 and 5 – Please change the color of the dot used on the Antarctica map to show the ice shelf location to something that stands out more clearly.

Section 4.3 – Is ERA5 not too coarse to resolve local katabatic wind effects?

L364 – Please include a citation after "relict lakes".

L370 – Specify which region you are referring to – DML or the grounded ice sheet?

L385 – 389 – Please quantify the "limited meltwater production compared to snow accumulation"

L406 – What "feedback mechanisms"? Larger differences in the extents of ponding compared with what?

L437 – "No evidence of subglacial drainage of lakes…". This statement is incorrect (Dunmire et al. 2020)

L440 – "Direct drainage into the ocean…" Are you referring to horizontal overflow drainage? If a lake drains vertically on an ice shelf it will likely drain to the ocean.

L444-445 – "r = ~0". Is there a figure for this? I have a hard time believing there is no correlation! From what region are you taking the near-surface temperature from? i.e. just areas where melt occurs or a larger area including upstream grounded ice or ocean?

L455 – "high-resolution climate modeling over Dronning Maud Land". What about RACMO (ie Lenaerts et. al 2017)?

L463 – Unnecessary to mention Fohn winds since they do not form in DML.

L483 – "Such shelves…" Which shelves? Do these shelves align with where melt is observed?

**Technical corrections**

L9 – "can cause firn air depletion" → "can potentially lead to firn air depletion"

L15 – move "ice shelves" after "Riiser Larsen, Nivlisen, and Roi Baudouin"

L17 – Please rephrase the sentence beginning with "Despite large interannual…" as it is a bit confusing

L21 – remove "in total, it"

L23 – Add ", ice-shelves…" after "Fimbulisen and Nivlisen"

L24 – Please change "the region" to "Dronning Maud Land" and "Dronning Maud Land" to "this region" in L25.

L35 – "destabilizing the ice sheet upstream" → "increasing upstream ice velocity"

L55 – "simpler" → "possible"

L56 – Add a comma after "clouds"

L127 – "pixel configuration" → "resolution"

L127 – "As in case of Landsat-8" → "Again,"

L160 – Add "However," before "Since Sentinel-2…"

L199 – Replace "near-surface" with "2 m" and remove ", measured 2 m above the ground" in the next line.

L206 – "melt seasons" →"melt season"

L223 – Add a comma after "size".

L233 – Remove "Judging"

L234 – Please reword this sentence to: "From the climate reanalysis data (Figure 2a), air temperature peaks around mid-January,…"

Table 2 caption: "ices helves" →"ice shelves"

L423 – Replace "draining" with "horizontally overflowing"

L475 – Move the Kuipers Munneke 2014 citation after "ice shelves" on the next line.

L482 – "liner" → "linear"

**References**

Banwell, A. F., MacAyeal, D. R., and Sergienko, O. V. (2013). Breakup of the Larsen B Ice Shelf triggered by chain reaction drainage of supraglacial lakes, *Geophys. Res. Lett.*, 40, 5872– 5876, doi:10.1002/2013GL057694.

Banwell, A., & Macayeal, D. (2015). Ice-shelf fracture due to viscoelastic flexure stress induced by fill/drain cycles of supraglacial lakes. *Antarctic Science,* 27 (6), 587-597. doi:10.1017/S0954102015000292.

Banwell, A. F., Datta, R. T., Dell, R. L., Moussavi, M., Brucker, L., Picard, G., Shuman, C. A., and Stevens, L. A. (2021). The 32-year record-high surface melt in 2019/2020 on the northern George VI Ice Shelf, Antarctic Peninsula, *The Cryosphere*, 15, 909–925, https://doi.org/10.5194/tc-15-909-2021.

Bell, R., Chu, W., Kingslake, J. *et al.* (2017). Antarctic ice shelf potentially stabilized by export of meltwater in surface river. *Nature,* **544**, 344–348. https://doi.org/10.1038/nature22048.

Dunmire, D., Lenaerts, J. T. M., Banwell, A. F., Wever, N., Shragge, J., & Lhermitte, S., et al. (2020). Observations of buried lake drainage on the Antarctic Ice Sheet. *Geophysical Research Letters*, 47, e2020GL087970. https://doi.org/10.1029/2020GL087970

Johnson, A., Hock, R., & Fahnestock, M. (2022). Spatial variability and regional trends of Antarctic ice shelf surface melt duration over 1979–2020 derived from passive microwave data. *Journal of Glaciology, 68*(269), 533-546. doi:10.1017/jog.2021.112.

Langley, E. S., Leeson, A. A., Stokes, C. R., and Jamieson, S. S. R. (2016), Seasonal evolution of supraglacial lakes on an East Antarctic outlet glacier, *Geophys. Res. Lett.*, 43, 8563–8571, doi:10.1002/2016GL069511.

Leeson, A. A., Forster, E., Rice, A., Gourmelen, N., & van Wessem, J. M. (2020). Evolution of supraglacial lakes on the Larsen B ice shelf in the decades before it collapsed. *Geophysical Research Letters*, 47, e2019GL085591. https://doi.org/10.1029/2019GL085591.

Lenaerts, J., Lhermitte, S., Drews, R. *et al.* (2017). Meltwater produced by wind–albedo interaction stored in an East Antarctic ice shelf. *Nature Clim Change* **7**, 58–62. https://doi.org/10.1038/nclimate3180.

Leppäranta, M., Järvinen, O., & Mattila, O. (2013). Structure and life cycle of supraglacial lakes in Dronning Maud Land. *Antarctic Science, 25*(3), 457-467. doi:10.1017/S0954102012001009.

Liston, G. E., Bruland, O, Winther, J, Elvehøy, H. & Sand, K (1999). Meltwater production in Antarctic blue-ice areas: sensitivity to changes in atmospheric forcing,Polar Research, 18:2, 283-290, DOI: 10.3402/polar.v18i2.6586.

Moussavi, M.; Pope, A.; Halberstadt, A.R.W.; Trusel, L.D.; Cioffi, L.; Abdalati, W. (2020). Antarctic Supraglacial Lake Detection Using Landsat 8 and Sentinel-2 Imagery: Towards Continental Generation of Lake Volumes. *Remote Sens.*, *12*, 134. https://doi.org/10.3390/rs12010134

Scambos, T., Fricker, H. E., Liu, C., Bohlander, J., Fastook, J., Sargent, A., Massom, R., and Wu, A. (2009). Ice shelf disintegration by plate bending and hydro-fracture: Satellite observations and

model results of the 2008 Wilkins ice shelf break-ups. *Earth and Planetary Science Letters* **280** (1-4), 51-61. https://doi.org/10.1016/j.epsl.2008.12.027

Stokes, C.R., Sanderson, J.E., Miles, B.W.J. *et al.* (2019). Widespread distribution of supraglacial lakes around the margin of the East Antarctic Ice Sheet. *Sci Rep* **9**, 13823. https://doi.org/10.1038/s41598-019-50343-5.

---

## Referee Comment (RC2)

**Review of tc-2023-4**
**Recent Evolution of Supraglacial Lakes on ice shelves in Dronning Maud Land, East Antarctica. Mahagaonkar et al. (2023)**

This paper uses a band thresholding approach to map surface meltwater over several ice shelves in Dronning Maud Land between 2014 and 2021. It is the first study to look at inter- **and** intra- annual variations in ponded water extent across several ice shelves in this region.

Overall, this paper is fairly well written, and adds some detail to our understanding of surface meltwater across Dronning Maud Land. However, in my opinion, the paper requires some major and minor edits prior to being ready for publication, which I have outlined below.

Major Comments

The authors use relatively new methods from Moussavi et al. (2020) to mask for clouds, rocks, and ocean prior to classifying water pixels. However, Moussavi et al. (2020) also provide a multi-band threshold approach for mapping surface meltwater, but the authors opt to just use an NDWI$_{ice}$ approach, with no justification for this decision. This is an odd decision to make, given the success of the methods provided by Moussavi et al. (2020), and given that the methodological approaches are now mixed. I would recommend mapping the meltwater using the methods provided by Moussavi et al. (2020), or giving a strong justification as to why this is not done.

In addition to this, please can I request that the authors provide the exact equations used for all methodological steps in their work. At times I find the methodological descriptions very vague.

Throughout the text, there is a significant lack of references to the relevant papers. Please can the authors comb through their work and make sure any idea that is not their own is referenced. This is particularly true for the discussion.

At present, there are a lot of interpretative and discursive statements in the results section, which need to be removed and placed into the discussion of the paper. The results should simply state the results presented by the study, and nothing more. Examples of this are parts (or all) of lines 292-293, 321-322, 336, 340-341.

The authors state numerous times that they see the transfer of water across the ice-shelf surface. However, I would argue that they present no evidence for this. The data collected here simply maps surface meltwater extent and does not prove flow or transfer of water in any dimension. It is not implausible that water is flowing across the ice-shelf surface along topographic gradients, but without evidence and analysis showing this, I think these inferences should be removed from the paper.

The discussion currently includes numerous claims that are neither supported by references from the literature nor from evidence in the paper. This discussion should build on phenomenon that the analysis finds and embed this in previously published literature.

Minor Comments

L8 (and elsewhere): The phrase ice sheet – ice shelf stability regime is a bit odd, perhaps rephrase to 'severe impacts on the stability of an ice shelf' and then weave in the subsequent implications for the ice sheets because of this.

L12: Change 'Antarctic ice sheet' to 'Antarctic Ice Sheet'. Check for this elsewhere too & check other nouns.

L27/28: This sentence is vague, try to add detail.

L30: If the water isn't being exported as run off, but surface melt is occurring, then tell the reader what the fate of that surface melt is.

L37: In this paragraph you start by talking about meltwater on ice shelves, but on this line you say meltwater can drain and cause glacier speed-up, which is something that would occur on grounded ice rather than an ice shelf. Please read through your paper again thoroughly and make sure you clearly separate out ideas pertaining to ice shelves vs. grounded ice.

L42: There are so many more references you could use here. If you just want to use a couple, then please write this as '(e.g. Kingslake et al., 2017; Stokes et al., 2019)'. This concept should be applied elsewhere throughout the paper too.

L46-48: The evolution of supraglacial lakes has also been observed on the Nivlisen Ice Shelf (Dell et al., 2020), and Roi Baudouin Ice Shelf (Dell et al., 2022). Both of these are in Dronning Maud Land.

L59: What do you mean by 'simple configuration', perhaps reword.

L66: The two papers you reference here actually use different methods, so which did you use?

L72: 'houses' is an odd choice of word, perhaps change.

L73: Change 'kms' to 'km'

L77: The periphery of what? This isn't very clear. Where exactly on the ice shelf are you referring to?

L80-83: Could you show this in a figure? Which glaciers feed which ice shelves?

L87: These papers aren't really Antarctic-wide. Find alterantives, depending on how you want to define Antarctic-wide.

L88-89: This study by Dell et al. (2020) should have been referenced much earlier on before you justify your study and your work. Also you should reference Dell et al. (2022) and Kinglake et al. (2015).

L92: Change 'Landsat-8' to Landsat 8

L104: Here and elsewhere, where did you download all your datasets from?

105: What is the common pixel size?

L104-105: Could this information be weaved into a different paragraph; it seems odd to have it structured in this way.

L109: It might be better to say that you use Collection 2 Landsat data before this point.

L111: Did you pick these bands because they were the bands used by Moussavi et al. (2020), if so then perhaps say this.

L114: Did you get a script for the per-pixel corrections from somewhere? If so perhaps reference it.

L146-147: What are the 'other outliers'?

L148: How much manual post processing was required? You could perhaps give an example of an image before and after manual post-processing as a supplementary figure.

L168: Reference needed.

L173: And Dell et al. (2020)

L182: What do you mean by ice-shelf region?

L191: What do you mean by 'Larger areas having such pixels were manually removed'? You removed lake edges found in Sentinel but not Landsat? Clarify.

L200: change 'temperature' to 'temperatures' (in two places). Change 'was' to 'were'.

L206: Change 'seasons' to 'season'.

L209: So do you chose to refer to areas or volumes?

L220: Do you have a figure showing what areas were randomly sampled? Did you randomly sample homogeneous features separately to homogenous features? Or did both types of features just get selected in the same random sample?

L225-227: What was the maximum difference, minimum difference etc? Does this justify your selection of a 1% relative uncertainty?

L232: Give the seasons in brackets after 'several seasons'

L256: This should probably be made clearer earlier in the text too, but did you look at every ice shelf in DML to start with, and then only find melt on five ice shelves? OR did you only look at these five ice shelves in the first place?

265: Here and elsewhere: What does 'viz' mean? Rewrite/ reword.

L272: Give examples of the years of low and high melt.

L285: remove '(within a particular melt season)'

L286: remove '(multiple melt seasons from different years)'

L339: I think you mean Fig 7.

L392-395: What are you trying to say here? It doesn't make much sense to me.

L396: Again, missing references, if you are just giving some then use (e.g. refs).

L423: Put these refs immediately after the ideas they are relevant for.

L440: Clarify what you mean by 'Direct surface drainage into the ocean'. I think you mean water flow off the ice shelf edge into the ocean, but this could also mean hydrofracture. Check this elsewhere too.

L451: Change 'Shackleton ice shelf' to 'Shackleton Ice Shelf'.
* * *
Figures

Figure 1: What dataset in Quantarctica did you use? It should have a reference as it would have come from somewhere.

Figure 3: Does the 10 Km scale apply to all the individual ice shelf maps? Give the inset map a scale.

Figure 6: For the 'b' figures, consider changing the Y axis to %'s, as this is how you present the data in the text . This Y axis also needs a label! X axis labels needed for both a and b!

Figure 7: Add X axis labels.

Table 2: Caption: correct 'ices helves'

Supplementary

Figure S3: Give the actual image information for imagery used. Apply this where appropriate elsewhere too.

Figure S6: Legend needed. If you are using REMA data it needs to be detailed in the methods of your paper.

Figure S7: Not referred to in the text? Also this builds heavily on the work of Moussavi et al. (2020), correct? This should be acknowledged. The figure also provides no info on the threshold values used.

A table of the satellite data used and data availability should be included as supplementary info.

References

Dell, R., Arnold, N., Willis, I., Banwell, A., Williamson, A., Pritchard, H. and Orr, A.: Lateral meltwater transfer across an Antarctic ice shelf, Cryosph., 14(7), 2313–2330, doi:10.5194/tc-14-2313-2020, 2020.

Dell, R. L., Banwell, A. F., Willis, I. C., Arnold, N. S., Halberstadt, A. R. W., Chudley, T. R. and Pritchard, H. D.: Supervised classification of slush and ponded water on Antarctic ice shelves using Landsat 8 imagery, J. Glaciol., 1–14, doi:10.1017/jog.2021.114, 2022.

Kingslake, J., Ng, F. and Sole, A.: Modelling channelized surface drainage of supraglacial lakes, J. Glaciol., 61(225), 185–199, doi:10.3189/2015JoG14J158, 2015.

Kingslake, J., Ely, J. C., Das, I. and Bell, R. E.: Widespread movement of meltwater onto and across Antarctic ice shelves, Nature, 544(7650), 349–352, doi:10.1038/nature22049, 2017.

Moussavi, M., Pope, A., Halberstadt, A. R. W., Trusel, L. D., Cioffi, L. and Abdalati, W.: Antarctic Supraglacial Lake Detection Using Landsat 8 and Sentinel-2 Imagery: Towards Continental Generation of Lake Volumes, Remote Sens., 12(1), 134, doi:10.3390/rs12010134, 2020.

Stokes, C. R., Sanderson, J. E., Miles, B. W. J., Jamieson, S. S. R. and Leeson, A. A.: Widespread distribution of supraglacial lakes around the margin of the East Antarctic Ice Sheet, Sci. Rep., 9(1), doi:10.1038/s41598-019-50343-5, 2019.

---

## Author Comment (AC1)

**Author's response to the comments from Reviewer #1**

Review of Mahagaonkar et al: Recent Evolution of Supraglacial Lakes on ice shelves in Dronning Maud Land, East Antarctica.

This study investigates the intra-seasonal and inter-annual evolution of supraglacial lakes in Dronning Maud Land, East Antarctica. The authors detect supraglacial lakes in Landsat-8 and Sentinel-2 satellite imagery from 2014-2021 for five ice shelf regions. They compare SGL area and volume with near-surface air temperature and positive degree days from the reanalysis model, ERA5 to climatologically explain SGL variability.

Overall, this manuscript is well-written and the figures are clear and informative. However, I have concerns about this paper's contribution to the field. Generally, it does not provide new information that is not already presented in other studies. Given the more specific comments listed below, I believe that major revisions are required.

*We thank the reviewer(s) for their efforts in reviewing the manuscript and providing constrictive feedback. We provide our responses to the specific comments below, including the concern about the novelty of the study (see Major Comments – Point 1).*

**Additionally, we have made some amendments to the sections pertaining to the relationship assessments between Supraglacial lakes and Climate, which is described below.**

*Initially, correlation calculations were performed using 'actual' SGL extents (SGL area and volume) and Mean DJF Temperatures. However, due to the large inherent differences in SGL extents between regions (e.g., Fimbulisen and Roi Baudouin East), the relatively good correlation for individual ice shelves was not reflected for the region as a whole. We have therefore normalized each annual maximum lake area/volume estimate with respect to their maximum for the whole study period. These normalized extents (values between 0 and 1) are now used to calculate correlations between lake extents and temperatures. This doesn't change the regional correlation values but increases the overall Dronning Maud Land correlation, better reflecting the relatively good local correlations. These values have been updated in Table 2 and relevant text has been updated as well (Section 3.7, Section 4.3, and Section 5.4). A figure has also been added to the supplement (Figure S9), showing a scatterplot of normalized areas with respect to mean summer temperatures.*

**Major comments**

1)      There are many previous studies that present surveys of supraglacial lakes for a specific region (Stokes et al., 2019 does so for all of East Antarctica, including DML). As such, this work feels incomplete without further investigating the climatic controls on supraglacial lake evolution in DML, especially since this is listed as an objective of this study (L65).

Further, the discussion section reads more as a literary review, providing little new information and insight into SGL evolution. Authors mention several times that the climatic factors that explain SGL development and evolution differences are still unknown and should be further investigated (i.e. L336, L359, L447, L471). I believe that this study should be expanded to provide additional insight into these climatic controls, especially for regions where air temperature and positive degree days do not explain SGL variability. For example, in L310 – what is the common factor controlling melting throughout the region? Similarly, in line 320, what are the local processes that also influence melting and ponding? This study feels incomplete without a further investigation into these climatic controls and local processes that impact meltwater ponding in DML.

*We agree that there are several studies reporting about SGLs in Dronning Maud Land, as part of their East Antarctic or Antarctic wide assessments. However, none of these have assessed long-term evolution of lakes on a multiyear basis covering the whole of Dronning Maud Land. The study by Dell et al. (2020) assesses the seasonal evolution of lakes over Nivlisen Ice Shelf for the 2016-2017 melt season only, whereas our study performs seasonal and interannual assessments over sizeable lakes of the entire Dronning Maud Land, found in five regions as discussed in the manuscript. Moreover, this study highlights the large variability of lakes within Dronning Maud Land and identifies years with similar relative extents (e.g., High – 2016-2017; Low – 2020-2021), which offer interesting insights into local and regional melting and ponding within Dronning Maud Land.*

*We think that the main result and novelty of this paper is the comprehensive information of SGL evolution in Dronning Maud Land between 2014 and 2021. This gives a good basis for further climatological analysis, and as a first step we have here investigated the basic relationships with surface air temperature. In a follow-up study, we are analyzing further climatic controls in more detail. Including such detailed analyses here would exceed space limitation of our already long manuscript. Therefore, we only briefly discuss other relevant processes in this manuscript, acknowledging their relevance and need for further analyses. Therefore, we think it would be best to not expand the current paper, but rather carry out further meteorological assessments (that includes several climatic components, e.g., solar radiation, wind, clouds etc.) in a separate, dedicated study.*

2)      The citations (especially in the introduction) are outdated and incomplete. Some specific examples are listed below:

- L30 – Is there a newer source that could be cited here? i.e. Johnson et al., 2022
  *Added (Johnson et al., 2022)*

- L34 – Banwell and MacAyeal 2015 is a more appropriate citation for ice shelf flexure and fracture.
  *Added (Banwell and MacAyeal, 2015)*
- L34 – Please replace the DeConto and Pollard 2015 citation with: Banwell et al., 2013 and Scambos et al., 2009
  *Replaced (DeConto and Pollard, 2015) with (Banwell et al., 2013; Scambos et al., 2009)*

- L35 - Please add a citation for "increased buttressing"
  *Added (Glasser et al., 2011)*

- L36 – This Kuipers-Munneke 2014 paper is on firn air depletion, not firn aquifers please cite (https://doi.org/10.1002/2013GL058389) instead.
  *Replaced with (Kuipers Munneke et al., 2014)*

- L41 – There are several newer studies that look at the pervasive meltwater ponding on the Antarctic Peninsula (Leeson et al., 2020, Banwell et al., 2021)
  *Added (Leeson et al., 2020; Banwell et al., 2021)*

- L46-48 - Langley et al., 2016 look at the seasonal evolution of supraglacial lakes on an outlet glacier in DML and should be mentioned here (also in L87-89).
  *Good point. We also added a note on the existence of small lakes around mountain nunataks: 'Langley et al.(2016) found presence of supraglacial lakes on Langhovde Glacier, just east of our study region, at elevations as high as 670 m a.s.l. Perennially ice-covered lakes exist even higher*

*at the edges of nunataks in the inland mountains of Dronning Maud Land (e.g., Faucher et al., 2019)'. The study of Langley et al. is now also cited later in the manuscript.*

*And added later: '… and Langley et al. (2016) presented a multiyear assessment of SGLs over Langhovde Glacier for years between 2000 and 2013'*

- Lenaerts et al., 2016 should be Lenaerts et al., 2017
  *Corrected all instances in the manuscript.*

- L53 – 60 – many citations are missing in this paragraph including: Leppäranta et al. 2013, Liston et al. 1999, Dunmire et al. 2020
  *Added Riiser Larsen (Leppäranta et al., 2013)*
  *Added (Dunmire et al., 2020)*
  *Added ' Using field observations from Jutulgryta (near Fimbulisen), Liston et al. (1999) developed a model to understand the sensitivity of meltwater production in blue-ice areas to atmospheric forcing, highlighting the importance of subsurface solar radiation and melting in snow and ice.'*

- L58 – Add Moussavi et al. 2020
  *Added (Moussavi et al., 2020)*

- L77 – Add Bell et al. 2017
  *Added (Bell et al., 2017)*

- L84 – Add Lenaerts et al. 2017
  *Added (Lenaerts et al., 2017)*

- Dunmire et al. 2020 should be cited in several places (L426 for "partial surficial freezing and subsequent insulation of deeper meltwater", L437-440 for DML lake drainage).
  *~L432: Added citation for 'partial surficial freezing and subsequent insulation of deeper meltwater' (Dunmire et al., 2020)*

  *~L442-443: Added – 'Modelling experiment by Dunmire et al., (2020) demonstrates the sensitivity of surface freezing to snow accumulation and highlights the possibility for buried non-frozen lakes throughout the cold austral winter.'*

  *~L447-451: Modified – 'Evidence of subglacial drainage of lakes though crevasses/cracks or englacial conduits in Dronning Maud Land is limited to one report by Dunmire et al. (2020) from the western part of Roi Baudouin, about 1km inland from the grounding line. Considering recent observations elsewhere in East Antarctica (e.g., over Amery Ice Shelf; Spergel et al., 2021 and Shackleton Ice Shelf; Arthur et al., 2020a), subglacial drainage could be a more important process than previously thought.'*

  *~L510: Deleted 'We did not find any evidence of englacial drainage into crevasses/fractures, nor any indications of direct drainage to the oceans over any of the study areas.'*

3)      Finally, there are several other satellite and in-situ observations that could be utilized to expand this analysis. I believe that using microwave imagery (e.g., from Sentinel-1) would help paint a more complete picture of DML lakes. For example, in L379, can this be investigated with microwave imagery?

Additionally, does the seasonal evolution pattern you observe throughout the melt season match backscatter signals from Sentinel-1 indicating surface melting (L399)? Because the seasonal evolution is a large part of this paper, I believe it would be beneficial to compare with Sentinel-1, a satellite with year-round frequent observations. Sentinel-1 observations could be further used to see when/if lakes freeze completely (L430-435).

*We agree that it could be beneficial to compare the observations from this study with Sentinel-1 data, especially over melt seasons where the optical coverage is not optimal. However, there are practically no high-resolution Sentinel-1 data (Interferometric Wide Swath – IW Mode) products over Dronning Maud Land available directly for use. Only Extra Wide Swath (EW) Mode data is available that is coarser in resolution compared to IW.  Also, it should be noted that Sentinel-1 data products are different in many ways to optical satellite images and would require completely different methodology for their processing and interpretation, which is beyond the scope of this work. Optical products have been used by several studies in the past to map and assess SGL evolution over Greenland (e.g., Williamson et al., 2018) and Antarctica (e.g., Arthur et al., 2020, 2022; Dell et al., 2020; Tuckett et al., 2021; Stokes et al., 2019; Moussavi et al., 2020) implying their suitability in this work. Since most seasons and regions have good coverage of optical data, we believe our seasonal evolution patterns are adequately represented.*

*Added the following sentence to Section 1: Introduction*

**"Additionally, the lack of high-resolution Sentinel-1 (Interferometric Wide Swath – IW Mode) data over Dronning Maud Land limits the possibility for their usage for large scale temporal assessments in the region."**

Further, extensive in-situ meteorological observations are available from Neumeyer station in DML. Why were these observations not utilized?

*Meteorological data from the Neumayer station, located on the Ekström Ice Shelf which is the second ice shelf to the east of Riiser Larsen, would not directly represent climatic conditions for any of the lake areas of this study as the station is closer to the coastal edge of the ice shelf, whereas most lakes are found near the grounding zones of the ice shelves. However, we acknowledge that these in-situ data are relevant and plan to assess it further in a follow-on study with more detailed meteorological analyses.*

**Minor comments**

L17 – Please specify which ice shelves showed "no significant meltwater lakes"
*Since there are several ice shelves with no significant ponding, they have been listed in the results section (4.1: Spatial Distribution of SGLs).*

L29 – Specify that runoff is not significant on Antarctica
*Specified – 'Antarctic meltwater runoff...'*

L49 – What do you mean by "major" ice shelves? Buttressing capacity? Size?
*Amended throughout the paper: 'major' to 'large'*

L59 – What do you mean by "simple configuration"?
*This sentence is now removed.*

L78 – 80 – No need to cite Trusel et al. 2013 twice in this same sentence.
*Removed (Trusel et al., 2013) citation once.*

L80 – 82: Please remove the sentence: "The ice shelves of Dronning… East Ragnhild Glaciers" as it is not necessary.
*Removed the mentioned sentence.*

L98 – 100: Please reword this sentence beginning with "We did not use the cloud detection…" as it is confusing.
*Rephrased the entire sequence to improve clarity. 'Filtration of scenes based on automatic cloud detection algorithms was avoided as it may also exclude useful scenes that have clear areas over lakes despite extensive clod cover elsewhere.'*

Why were the 5 ice shelves used in this study chosen over other ice shelves in DML?
*The study was not restricted to the 5 ice shelves, but covers the entire Dronning Maud Land, where sizeable lakes are only found near these 5 ice shelves. This has now been clarified in several places in the manuscript: Abstract ('all ice shelves'), Section 3.1 on manually identifying areas with lakes. Section 4.1: 'No significant meltwater ponding was observed on other ice shelves in Dronning Maud Land'.*

Sections 3.2-3.3: It is unclear to me where you followed methods from other papers and where you did not. For example, in lines 117-22, was this still following Moussavi et al 2020 or did you use different thresholds? If the method is the same as work previously published, then the text can be simplified greatly. If not, please explain why you chose not to follow previously established methods. Also, are there any periods for which data is lacking? If so, please explain.
*Section 3 has been entirely rewritten and simplified. References to Moussavi et al. (2020) have been added where applicable, and differences have been explained better.*

L141 – Why do you use a threshold to exclude shallow lakes? I would think that by excluding shallow lakes you lead to an underestimation of SGLs as shallow lakes are still lakes!
*Added – 'to avoid overestimation of SGLs (Arthur et al., 2020; Dell et al., 2020) due to misclassification of slush and insignificant water pockets.'.*

*This threshold is also used by several other studies over Antarctica (e.g., Arthur et al., 2020; Dell et al., 2020) and Greenland (e.g., Williamson et al., 2018) for similar reasons.*

Section 3.4 – It is my understanding that the estimation of lake depth and volume is largely based off previously published methods? If so, this section can be greatly reduced.
*Agreed. The Section 3 has been entirely rewritten and simplified.*

Figure S1 – Why does Landsat 8 estimate greater depth for deeper lakes?

*We have not been able to identify why Landsat 8 appears to estimate higher depths than Sentinel-2, however we added an explanation for how the differences could arise.*

*Section 3.6: 'However, depths estimated by Landsat 8 appear to be slightly higher than Sentinel-2 estimates. Such differences could arise from cloud adjacency effect (Pope et al., 2016; Williamson et al., 2018) or due to the difference in their original spatial and spectral resolutions.'*

*Figure S1 Text - 'The difference between Landsat-8 and Sentinel-2 depths (Mean Bias = 0.08 m, RMSE ~ 0.21) could be due to cloud adjacency effect (Pope et al., 2016; Williamson et al., 2018) or due to the difference in their original spatial and spectral resolutions. However, the depths estimated by Sentinel-2 fall within the error range of Landsat-8 estimated depths (0.28 m. for red band; Pope et al., 2016).'*

Section 3.6 – why do you use ERA5 here?
*Added: 'Due to absence of any long-term in-situ meteorological observations covering the expanse of the study area, we use ERA5 climate reanalysis data'*

L219 – How were the manually digitized lake boundaries created? Were they created completely independently from the automatic lake masks?
*Yes, the manual digitization was carried out independently of the automatically generated lake masks. Landsat-8/Sentinel-2 scenes were used as a background while mapping lake boundaries.*

L228 – I believe that uncertainty with depth calculations should be considered as well in the uncertainty range for lake volume.
*Agreed. Added '…a depth uncertainty of 0.21 m determined from the RMSE difference between separate Landsat 8 and Sentinel-2 depth estimates.'.*

L240 – 250 – This section is a bit confusing to me and I think could be reworded for simplicity.
*We have made an attempt to simplify.*

Table 1 – What is the vertical curve on the left-hand side of the table (over the ponding/advection phases column)?
*The curve has been removed.*

L 258 – Refer to Figure 3 for the different ice shelf regions.
*Added (Figure 3) at the end of the sentence.*

L265 – What do you mean by "viz."? (also L507)
*The abbreviation 'viz.' has been replaced with 'namely'.*

L266 – Specify what are the years with high melting.
*Added: '(2016-2017 and 2019-2020)'*

L268-269 – Please include error when quantifying lake depths.
*Added error range in all instances.*

Figure 3 – I think it would be nice to also include the blue ice areas in this figure.
*Updated the figure by including blue ice areas and a legend entry indicating blue ice areas.*

Line 280 – More specifically, what is different about the Fimbulisen/Muninisen firn pack? Less blue ice? More firn air content?
*Due to lack of high-resolution Firn Air Content (FAC) dataset, we are unable to state what the differences in FAC over different regions of DML could be. Looking at the 27 km FAC product from IMAU-FDM Model, the differences seem to be insignificant over our areas of interest, however we are not confident due to the resolution of the product. Due to this lack of knowledge, we removed the statement '…, likely due to percolation into the unsaturated firn pack that surrounds the meltwater ponds.'*

Figures 4 and 5 – Please change the color of the dot used on the Antarctica map to show the ice shelf location to something that stands out more clearly.
*Changed the color of the dot to red in both images (and in Figure S4 and Figure S5)*

Section 4.3 – Is ERA5 not too coarse to resolve local katabatic wind effects?
*We agree that ERA5 (~31 km) is too coarse to resolve katabatic winds. This limits us from using ERA5 products in our further meteorological assessments; for which we are working closely with MAR Group for high resolution climate products ~5.5kms.*

L364 – Please include a citation after "relict lakes".
*Added citation – (Lenaerts et al., 2017)*

L370 – Specify which region you are referring to – DML or the grounded ice sheet?
*Specified – grounding zone, primarily above the grounding line.*

L385 – 389 – Please quantify the "limited meltwater production compared to snow accumulation"
*Rephrased to "a relatively small meltwater production compared to snow accumulation (having melt-over-accumulation ratio < 0.7; van Wessem et al., 2023; Pfeffer et al., 1991)"*

L406 – What "feedback mechanisms"? Larger differences in the extents of ponding compared with what?
*Added 'snow-ice-albedo feedback mechanisms'*
*Modified second sentence to 'larger extents of ponding compared to the average.'*

L437 – "No evidence of subglacial drainage of lakes…". This statement is incorrect (Dunmire et al. 2020)
*Modified to add reports of Dunmire et al. (2020).*
*"Evidence of subglacial drainage of lakes though crevasses/cracks or englacial conduits in Dronning Maud Land is limited to one report by Dunmire et al. (2020) from the western part of Roi Baudouin, about 1km inland from the grounding line."*

L440 – "Direct drainage into the ocean…" Are you referring to horizontal overflow drainage? If a lake drains vertically on an ice shelf it will likely drain to the ocean.
*Modified to 'Direct horizontal overflow (runoff over the calving edge of the shelves)'*

L444-445 – "r = ~0". Is there a figure for this? I have a hard time believing there is no correlation! From what region are you taking the near-surface temperature from? i.e. just areas where melt occurs or a larger area including upstream grounded ice or ocean?
*Thanks for pointing this out. After the changes made to the way correlation is calculated (by Normalizing SGL extents; explained at beginning of this document), the DML wide correlation indeed increases. This is also presented in the new Figure S9.*

*The ERA5 grid points from where the near-surface temperature is taken are presented in Figure S2. As explained in the revised text in Section 3.7, these points are chosen to best represent the location of origin of SGLs (typically the area near the grounding line).*

L455 – "high-resolution climate modeling over Dronning Maud Land". What about RACMO (ie Lenaerts et. al 2017)?
*We were informed by the RACMO group that currently, RACMO2.3p2 products are only available at 27 km resolution, like that of ERA5, for Dronning Maud Land. Higher resolution products that were generated were found to have a melt bias and precipitation bias due to which the simulations were not extended/continued. Therefore, not included here.*

L463 – Unnecessary to mention Fohn winds since they do not form in DML.
*Deleted respective sentences.*

L483 – "Such shelves…" Which shelves? Do these shelves align with where melt is observed?
*Amended to 'Such vulnerable shelves are also present in Dronning Maud Land (Figure 4 from Lai et al. (2020)) and coincide with the regions where melt ponding is observed, highlighting the importance… '*

**Technical corrections**

L9 – "can cause firn air depletion" ⮕ "can potentially lead to firn air depletion"
*Modified.*

L15 – move "ice shelves" after "Riiser Larsen, Nivlisen, and Roi Baudouin"
*Moved.*

L17 – Please rephrase the sentence beginning with "Despite large interannual…" as it is a bit confusing
*Deleted the sentence.*

L21 – remove "in total, it"
*Removed.*

L23 – Add ", ice-shelves…" after "Fimbulisen and Nivlisen"
*Added 'areas' instead, as melt ponds in Fimbulisen area are not exactly on the ice shelf but over the grounded ice.*

L24 – Please change "the region" to "Dronning Maud Land" and "Dronning Maud Land" to "this region" in L25.
*Amended.*

L35 – "destabilizing the ice sheet upstream" ⮕ "increasing upstream ice velocity"
*Amended.*

L55 – "simpler" ⮕ "possible"
*Amended.*

L56 – Add a comma after "clouds"
*Added.*

L127 – "pixel configuration" ⮕ "resolution"
*Changed.*

L127 – "As in case of Landsat-8" ⮕ "Again,"
*Changed.*

L160 – Add "However," before "Since Sentinel-2…"
*Added.*

L199 – Replace "near-surface" with "2 m" and remove ", measured 2 m above the ground" in the next line.
*Replaced, also added 'and Positive Degree Days (PDD)'.*

L206 – "melt seasons" ⮕ "melt season"
*Changed.*

L223 – Add a comma after "size".
*Added.*

L233 – Remove "Judging"
*Removed.*

L234 – Please reword this sentence to: "From the climate reanalysis data (Figure 2a), air temperature peaks around mid-January,…"
*Modified.*

Table 2 caption: "ices helves" → "ice shelves"
*Corrected.*

L423 – Replace "draining" with "horizontally overflowing"
*Replaced.*

L475 – Move the Kuipers Munneke 2014 citation after "ice shelves" on the next line.
*Moved.*

L482 – "liner" → "linear"
*Corrected.*

**References**

Arthur, J. F., Stokes, C. R., Jamieson, S. S. R., Carr, J. R., and Leeson, A. A.: Distribution and seasonal evolution of supraglacial lakes on Shackleton Ice Shelf, East Antarctica, Cryosph., 14, 4103–4120, https://doi.org/10.5194/tc-14-4103-2020, 2020.

Arthur, J. F., Stokes, C. R., Jamieson, S. S. R., Rachel Carr, J., Leeson, A. A., and Verjans, V.: Large interannual variability in supraglacial lakes around East Antarctica, Nat. Commun., 13, https://doi.org/10.1038/s41467-022-29385-3, 2022.

Banwell, A. F. and MacAyeal, D. R.: Ice-shelf fracture due to viscoelastic flexure stress induced by fill/drain cycles of supraglacial lakes, Antarct. Sci., 27, 587–597, https://doi.org/10.1017/S0954102015000292, 2015.

Banwell, A. F., MacAyeal, D. R., and Sergienko, O. V.: Breakup of the Larsen B Ice Shelf triggered by chain reaction drainage of supraglacial lakes, Geophys. Res. Lett., 40, 5872–5876, https://doi.org/10.1002/2013GL057694, 2013.

Banwell, A. F., Datta, R. T., Dell, R., Moussavi, M., Brucker, L., Picard, G., Shuman, C. A., and Stevens, L. A.: The 32-year record-high surface melt in 2019/2020 on the northern George VI Ice Shelf, Antarctic Peninsula, Cryosph., 15, 909–925, https://doi.org/10.5194/tc-15-909-2021, 2021.

Bell, R. E., Chu, W., Kingslake, J., Das, I., Tedesco, M., Tinto, K. J., Zappa, C. J., Frezzotti, M., Boghosian, A., and Lee, W. S.: Antarctic ice shelf potentially stabilized by export of meltwater in surface river, Nature, 544, 344–348, https://doi.org/10.1038/nature22048, 2017.

Dell, R., Arnold, N., Willis, I., Banwell, A. F., Williamson, A., Pritchard, H. D., and Orr, A.: Lateral meltwater transfer across an Antarctic ice shelf, Cryosphere, 14, 2313–2330, https://doi.org/10.5194/tc-14-2313-2020, 2020.

Dunmire, D., Lenaerts, J. T. M., Banwell, A. F., Wever, N., Shragge, J., Lhermitte, S., Drews, R., Pattyn, F., Hansen, J. S. S., Willis, I. C., Miller, J., and Keenan, E.: Observations of Buried Lake Drainage on the Antarctic Ice Sheet, Geophys. Res. Lett., 47, https://doi.org/10.1029/2020GL087970, 2020.

Faucher, B., Lacelle, D., Fisher, D. A., Andersen, D. T., and McKay, C. P.: Energy and water mass balance of Lake Untersee and its perennial ice cover, East Antarctica, Antarct. Sci., 31, 271–285, https://doi.org/10.1017/S0954102019000270, 2019.

Glasser, N. F., Scambos, T. A., Bohlander, J., Truffer, M., Pettit, E., and Davies, B. J.: From ice-shelf tributary to tidewater glacier: continued rapid recession, acceleration and thinning of Röhss Glacier following the 1995 collapse of the Prince Gustav Ice Shelf, Antarctic Peninsula, J. Glaciol., 57, 397–406,

https://doi.org/10.3189/002214311796905578, 2011.

Johnson, A., Hock, R., and Fahnestock, M.: Spatial variability and regional trends of Antarctic ice shelf surface melt duration over 1979–2020 derived from passive microwave data, J. Glaciol., 68, 533–546, https://doi.org/10.1017/jog.2021.112, 2022.

Kuipers Munneke, P., M. Ligtenberg, S. R., van den Broeke, M. R., van Angelen, J. H., and Forster, R. R.: Explaining the presence of perennial liquid water bodies in the firn of the Greenland Ice Sheet, Geophys. Res. Lett., 41, 476–483, https://doi.org/10.1002/2013GL058389, 2014.

Lai, C. Y., Kingslake, J., Wearing, M. G., Chen, P. H. C., Gentine, P., Li, H., Spergel, J. J., and van Wessem, J. M.: Vulnerability of Antarctica's ice shelves to meltwater-driven fracture, Nature, 584, 574–578, https://doi.org/10.1038/s41586-020-2627-8, 2020.

Langley, E. S., Leeson, A. A., Stokes, C. R., and Jamieson, S. S. R.: Seasonal evolution of supraglacial lakes on an East Antarctic outlet glacier, Geophys. Res. Lett., 43, 8563–8571, https://doi.org/10.1002/2016GL069511, 2016.

Leeson, A. A., Forster, E., Rice, A., Gourmelen, N., and Wessem, J. M.: Evolution of Supraglacial Lakes on the Larsen B Ice Shelf in the Decades Before it Collapsed, Geophys. Res. Lett., 47, https://doi.org/10.1029/2019GL085591, 2020.

Lenaerts, J. T. M., Lhermitte, S., Drews, R., Ligtenberg, S. R. M., Berger, S., Helm, V., Smeets, C. J. P. P., Broeke, M. R. van den, van de Berg, W. J., van Meijgaard, E., Eijkelboom, M., Eisen, O., and Pattyn, F.: Meltwater produced by wind–albedo interaction stored in an East Antarctic ice shelf, Nat. Clim. Chang., 7, 58–62, https://doi.org/10.1038/nclimate3180, 2017.

Leppäranta, M., Järvinen, O., and Mattila, O.-P.: Structure and life cycle of supraglacial lakes in Dronning Maud Land, Antarct. Sci., 25, 457–467, https://doi.org/10.1017/S0954102012001009, 2013.

Liston, G. E., Bruland, O., Winther, J.-G., Elvehøy, H., and Sand, K.: Meltwater production in Antarctic blue-ice areas: sensitivity to changes in atmospheric forcing, Polar Res., 18, 283–290, https://doi.org/10.3402/polar.v18i2.6586, 1999.

Moussavi, M., Pope, A., Halberstadt, A. R. W., Trusel, L. D., Cioffi, L., and Abdalati, W.: Antarctic Supraglacial Lake Detection Using Landsat 8 and Sentinel-2 Imagery: Towards Continental Generation of Lake Volumes, Remote Sens., 12, 134, https://doi.org/10.3390/rs12010134, 2020.

Pfeffer, W. T., Meier, M. F., and Illangasekare, T. H.: Retention of Greenland runoff by refreezing: Implications for projected future sea level change, J. Geophys. Res., 96, 22117, https://doi.org/10.1029/91JC02502, 1991.

Pope, A., Scambos, T. A., Moussavi, M., Tedesco, M., Willis, M., Shean, D., and Grigsby, S.: Estimating supraglacial lake depth in West Greenland using Landsat 8 and comparison with other multispectral methods, Cryosphere, 10, 15–27, https://doi.org/10.5194/tc-10-15-2016, 2016.

Scambos, T. A., Fricker, H. A., Liu, C. C., Bohlander, J., Fastook, J., Sargent, A., Massom, R., and Wu, A. M.: Ice shelf disintegration by plate bending and hydro-fracture: Satellite observations and model results of the 2008 Wilkins ice shelf break-ups, Earth Planet. Sci. Lett., 280, 51–60, https://doi.org/10.1016/j.epsl.2008.12.027, 2009.

Spergel, J. J., Kingslake, J., Creyts, T., van Wessem, M., and Fricker, H. A.: Surface meltwater drainage and ponding on Amery Ice Shelf, East Antarctica, 1973–2019, J. Glaciol., 67, 985–998,

https://doi.org/10.1017/jog.2021.46, 2021.

Stokes, C. R., Sanderson, J. E., Miles, B. W. J., Jamieson, S. S. R., and Leeson, A. A.: Widespread distribution of supraglacial lakes around the margin of the East Antarctic Ice Sheet, Sci. Rep., 9, 1–14, https://doi.org/10.1038/s41598-019-50343-5, 2019.

Tuckett, P. A., Ely, J. C., Sole, A. J., Lea, J. M., Livingstone, S. J., Jones, J. M., and Van Wessem, J. M.: Automated mapping of the seasonal evolution of surface meltwater and its links to climate on the Amery Ice Shelf, Antarctica, Cryosphere, 15, 5785–5804, https://doi.org/10.5194/tc-15-5785-2021, 2021.

van Wessem, J. M., van den Broeke, M. R., Wouters, B., and Lhermitte, S.: Variable temperature thresholds of melt pond formation on Antarctic ice shelves, Nat. Clim. Chang., 13, 161–166, https://doi.org/10.1038/s41558-022-01577-1, 2023.

Williamson, A. G., Banwell, A. F., Willis, I. C., and Arnold, N. S.: Dual-satellite (Sentinel-2 and Landsat 8) remote sensing of supraglacial lakes in Greenland, Cryosphere, 12, 3045–3065, https://doi.org/10.5194/tc-12-3045-2018, 2018.

---

## Author Comment (AC2)

**Author's response to the comments from Reviewer #2**

This paper uses a band thresholding approach to map surface meltwater over several ice shelves in Dronning Maud Land between 2014 and 2021. It is the first study to look at inter- and intra- annual variations in ponded water extent across several ice shelves in this region.

Overall, this paper is fairly well written, and adds some detail to our understanding of surface meltwater across Dronning Maud Land. However, in my opinion, the paper requires some major and minor edits prior to being ready for publication, which I have outlined below.

*We would like to thank the reviewer(s) for their work on our manuscript and the encouraging comments. We address each of the suggestions and provide responses to specific comments below.*

***Additionally, we have made some amendments to the sections pertaining to the relationship assessments between Supraglacial lakes and Climate, which is described below.***

*Initially, correlation calculations were performed using 'actual' SGL extents (SGL area and volume) and Mean DJF Temperatures. However, due to the large inherent differences in SGL extents between regions (e.g., Fimbulisen and Roi Baudouin East), the relatively good correlation for individual ice shelves was not reflected for the region as a whole. We have therefore normalized each annual maximum lake area/volume estimate with respect to their maximum for the whole study period. These normalized extents (values between 0 and 1) are now used to calculate correlations between lake extents and temperatures. This doesn't change the regional correlation values but increases the overall Dronning Maud Land correlation, better reflecting the relatively good local correlations. These values have been updated in Table 2 and relevant text has been updated as well (Section 3.7, Section 4.3, and Section 5.4). A figure has also been added to the supplement (Figure S9), showing a scatterplot of normalized areas with respect to mean summer temperatures.*

**Major Comments**

The authors use relatively new methods from Moussavi et al. (2020) to mask for clouds, rocks, and ocean prior to classifying water pixels. However, Moussavi et al. (2020) also provide a multi- band threshold approach for mapping surface meltwater, but the authors opt to just use an NDWIice approach, with no justification for this decision. This is an odd decision to make, given the success of the methods provided by Moussavi et al. (2020), and given that the methodological approaches are now mixed. I would recommend mapping the meltwater using the methods provided by Moussavi et al. (2020), or giving a strong justification as to why this is not done.
*Clarified the text by rephrasing the methodology (Section3: 3.1 – 3.4). The mapping of lakes was done using all steps recommended by Moussavi et al. (2020).*

In addition to this, please can I request that the authors provide the exact equations used for all methodological steps in their work. At times I find the methodological descriptions very vague.
*To address this, the methodology section has been rephrased entirely and a methodological flowchart has been added as supplementary Figure S8, including band indices and thresholds used. Additionally, the equation of NDSI has been added in the manuscript, and all the equations have been presented in Figure S8.*

Throughout the text, there is a significant lack of references to the relevant papers. Please can the authors comb through their work and make sure any idea that is not their own is referenced. This is particularly true for the discussion.
*Added references at several places in the manuscript. Some examples:*

*~ L474 –' …the surface meltwater has drained or begins to freeze at the surface to form relict lakes (as observed by Lenaerts et al. (2017)'*
*~L480 – ' …feedback mechanism of exposed blue ice areas that enhances melting of surrounding ice (Bell et al., 2018; Lenaerts et al., 2017).'*
*~ L484 – 486 – '… laterally transported across the grounding zone and over the ice shelves by surface streams and channels, as also observed by (Dell et al., (2020) over Nivlisen Ice Shelf.'*
*~ L545 – '…directly draining into the ocean through surface streams (Bell et al. 2017).'.*
*~L547 –'… partial surficial freezing and subsequent insulation of deeper meltwater (Dunmire et al., 2020)…'*

*Please see also the response to reviewer 1 on adding further references in the manuscript.*

At present, there are a lot of interpretative and discursive statements in the results section, which need to be removed and placed into the discussion of the paper. The results should simply state the results presented by the study, and nothing more. Examples of this are parts (or all) of lines 292-293, 321-322, 336, 340-341.
*We agree and have removed several discussion-type statements:*
*~L292-293 Removed – 'where meltwater production is more efficient due to the exposed blue ice having lower albedo.'*
*~L321-322 Removed – 'This mixed pattern in nearby regions suggests that local processes also influence melting and ponding in the region.'*
*~L336 Removed – 'The reasons for these large variations in surface ponding over different regions (over 2 orders of magnitude) remain unexplained.'*
*~L340-341 Removed – 'This relative consistency in melt extents for different areas indicates that there are common climatic factors that control melting and ponding in Dronning Maud Land at large'.*
*~L282 Removed – 'presumably due to the low supply of meltwater'*
*~L284 Removed – 'likely due to percolation into the unsaturated firn pack that surrounds the meltwater ponds.'*

The authors state numerous times that they see the transfer of water across the ice-shelf surface. However, I would argue that they present no evidence for this. The data collected here simply maps surface meltwater extent and does not prove flow or transfer of water in any dimension. It is not implausible that water is flowing across the ice-shelf surface along topographic gradients, but without evidence and analysis showing this, I think these inferences should be removed from the paper.
*Evidence for Lateral transfer of meltwater is presented in Figure 4 and Figure 5 of the manuscript (in addition to Figures S4 and S5 of the supplement), and has been interpreted similarly by earlier studies (e.g.,* Dell et al., 2020; Kingslake et al., 2017)*). We noticed that nothing about lateral transfer was mentioned in the figure caption - this has now been added. Additionally, we have also added Supplementary Animation 1 as supporting evidence of meltwater transfer.*

*Since the transfer of meltwater is also reported by Dell at al. (2020) over Nivlisen Ice Shelf, we have referenced this following the discussion of meltwater transfer. E.g., L377 – 378 : 'The produced meltwater is laterally transported across the grounding zone and over the ice shelves by surface streams and channels, as also observed by Dell et al. (2020) over Nivlisen Ice Shelf.'*

The discussion currently includes numerous claims that are neither supported by references from the literature nor from evidence in the paper. This discussion should build on phenomenon that the analysis finds and embed this in previously published literature.
*Thanks for pointing this out. We have removed several such statements from the manuscript.*

**Minor Comments**

L8 (and elsewhere): The phrase ice sheet – ice shelf stability regime is a bit odd, perhaps rephrase to 'severe impacts on the stability of an ice shelf' and then weave in the subsequent implications for the ice sheets because of this.
*Modified accordingly at two instances on ~L8 and ~L40.*

L12: Change 'Antarctic ice sheet' to 'Antarctic Ice Sheet'. Check for this elsewhere too & check other nouns.
*Corrected at several instances.*

L27/28: This sentence is vague, try to add detail.
*Replaced the sentence with some details.*
*'Ice shelves are floating extensions of Antarctic Ice Sheet and play a critical role in regulating the flow of ice from the continent into the ocean. As the climate warms, these ice shelves are becoming increasingly vulnerable to melting and collapse, which can accelerate the discharge of ice from the grounded ice sheet into the ocean and contribute to sea level rise'.*

L30: If the water isn't being exported as run off, but surface melt is occurring, then tell the reader what the fate of that surface melt is.
*Modified the following sentence to include the fate of meltwater*
*'Large amount of the meltwater produced due to melting of snow and ice (Echelmeyer et al., 1991) percolates and refreezes in the underlying layer of firn, whereas the remaining water ponds in topographical depressions, typically occurring around grounding zones (Stokes et al., 2019), leading to formation of supraglacial lakes.'*

L37: In this paragraph you start by talking about meltwater on ice shelves, but on this line you say meltwater can drain and cause glacier speed-up, which is something that would occur on grounded ice rather than an ice shelf. Please read through your paper again thoroughly and make sure you clearly separate out ideas pertaining to ice shelves vs. grounded ice.
*Removed the statement.*

L42: There are so many more references you could use here. If you just want to use a couple, then please write this as '(e.g., Kingslake et al., 2017; Stokes et al., 2019)'. This concept should be applied elsewhere throughout the paper too.
*Agreed and modified accordingly. See also the response to reviewer 1 on adding further references in the manuscript.*

L46-48: The evolution of supraglacial lakes has also been observed on the Nivlisen Ice Shelf (Dell et al., 2020), and Roi Baudouin Ice Shelf (Dell et al., 2022). Both of these are in Dronning Maud Land.
*Added details of Dell et al., 2020 in the paragraph.*

L59: What do you mean by 'simple configuration', perhaps reword.
*Sentence removed.*

L66: The two papers you reference here actually use different methods, so which did you use?
*Removed the incorrect reference (Williamson et al., 2018).*

L72: 'houses' is an odd choice of word, perhaps change.
*Changed to 'has'.*

L73: Change 'kms' to 'km'
*Corrected.*

L77: The periphery of what? This isn't very clear. Where exactly on the ice shelf are you referring to?
*Removed the sentence as the context in which it was initially written is lost.*

L80-83: Could you show this in a figure? Which glaciers feed which ice shelves?
*This sentence has been removed on suggestion of Reviewer # 1. The main glaciers feeding the ice shelves are indicated Figure 1.*

L87: These papers aren't really Antarctic-wide. Find alterantives, depending on how you want to define Antarctic-wide.
*Changed 'Antarctic-wide studies' to 'Various studies'.*

L88-89: This study by Dell et al. (2020) should have been referenced much earlier on before you justify your study and your work. Also you should reference Dell et al. (2022) and Kinglake et al. (2015).
*Added references to Dell et al., 2020 in the Introduction.*
*Added references to Dell et al. (2022)and Kingslake et al. (2015)in the paragraph.*

L92: Change 'Landsat-8' to Landsat 8
*Changed several instances in the manuscript.*

L104: Here and elsewhere, where did you download all your datasets from?
*Added links to the websites from where the data was downloaded. Also, the access information to all the data that is used in his work is detailed in **'Code and data availability'** section after 'Conclusion'.*

105: What is the common pixel size?
*Addressed in the rephrased methodology section.*

L104-105: Could this information be weaved into a different paragraph; it seems odd to have it structured in this way.
*Removed the sentences.*

L109: It might be better to say that you use Collection 2 Landsat data before this point.
*Addressed in the rephrased methodology section.*

L111: Did you pick these bands because they were the bands used by Moussavi et al. (2020), if so then perhaps say this.
*Addressed in the rephrased methodology section.*

L114: Did you get a script for the per-pixel corrections from somewhere? If so perhaps reference it.
*Added: (Williamson et al. (2018); script by Neil Arnold, University of Cambridge).*

L146-147: What are the 'other outliers'?
*Changed 'other outliers' → 'Outliers (e.g. exposed rock, insignificant water pockets, shadows and clouds)'*

L148: How much manual post processing was required? You could perhaps give an example of an image before and after manual post-processing as a supplementary figure.
*Added figure (Figure S7) in the supplement showing the occasional need for manual post processing.*

L168: Reference needed.
*Added reference (Pope et al., 2016).*

L173: And Dell et al. (2020)
*Added reference.*

L182: What do you mean by ice-shelf region?
*Since lakes were not only present on ice shelves but also on grounded ice (e.g., Fimbulisen and Muninisen) we decided to say 'ice-shelf regions', not just 'ice-shelf').*

L191: What do you mean by 'Larger areas having such pixels were manually removed'? You removed lake edges found in Sentinel but not Landsat? Clarify.
*Since the sentence was misleading and incorrect, we have removed the sentence. The elimination in Sentinel-2 outputs was done only to identify common pixels for Landsat 8 – Sentinel-2 comparison and this is understood in the next sentence '..we first identified common water pixels..'.*

L200: change 'temperature' to 'temperatures' (in two places). Change 'was' to 'were'.
*Corrected.*

L206: Change 'seasons' to 'season'.
*Corrected.*

L209: So do you chose to refer to areas or volumes?
*Modified to refer only to correlation values with respect to maximum lake areas.*

L220: Do you have a figure showing what areas were randomly sampled? Did you randomly sample homogeneous features separately to homogenous features? Or did both types of features just get selected in the same random sample?
*We randomly chose areas bearing in mind to contain both homogenous and inhomogeneous lakes. The text has been changed to better reflect this:*
*"…four randomly selected areas that contain water bodies of both homogenous and inhomogeneous character…"*

L225-227: What was the maximum difference, minimum difference etc? Does this justify your selection of a 1% relative uncertainty? Error is a function of lake size.
*Maximum difference is 0.94 km$^2$ (Area: 4.22 km$^2$) and Minimum difference is 0.07 km$^2$ (Area: 0.055 km$^2$). Although there are certain cases where percentage of difference is as high as ~400% (mostly in very small sized lakes), we can say that on average 1% error is justifiable, as these small lakes would have minimal contribution to the cumulative lake area.*

L232: Give the seasons in brackets after 'several seasons'
*Added 2 examples.*

L256: This should probably be made clearer earlier in the text too, but did you look at every ice shelf in DML to start with, and then only find melt on five ice shelves? OR did you only look at these five ice shelves in the first place?
*Agreed. We have added a sub-section to Section 3: Data and Methods:*

*'3.1 Identifying areas with SGLs*
*To identify locations with sizeable SGLs in Dronning Maud Land, we closely assessed ~120 Landsat 8 scenes captured in January 2017 and January 2021 (time periods randomly selected), representing the peak austral summer seasons. Careful visual inspection of these scenes was carried out to outline areas with possible SGL occurrence. After identification of these areas, further assessments were restricted to scenes covering these areas only.'*

265: Here and elsewhere: What does 'viz' mean? Rewrite/ reword.
*Replace 'viz' with 'namely'.*

L272: Give examples of the years of low and high melt.
*Added one example each.*

L285: remove '(within a particular melt season)'
*Removed.*

L286: remove '(multiple melt seasons from different years)'
*Removed.*

L339: I think you mean Fig 7.
*Corrected.*

L392-395: What are you trying to say here? It doesn't make much sense to me.
*Rephrased the sentences.*

L396: Again, missing references, if you are just giving some then use (e.g. refs).
*Added 'e.g.,'.*

L423: Put these refs immediately after the ideas they are relevant for.
*Moved references accordingly.*

L440: Clarify what you mean by 'Direct surface drainage into the ocean'. I think you mean water flow off the ice shelf edge into the ocean, but this could also mean hydrofracture. Check this elsewhere too.
*Corrected → Direct Horizontal overflow (runoff over the frontal edge of the shelves)*

L451: Change 'Shackleton ice shelf' to 'Shackleton Ice Shelf'.
*Corrected.*

**Figures**

Figure 1: What dataset in Quantarctica did you use? It should have a reference as it would have come from somewhere.
*Added source: SCAR's Antarctic Digital Database (ADD) through Quantarctica.*

Figure 3: Does the 10 Km scale apply to all the individual ice shelf maps? Give the inset map a scale.
*Changed 'Scale' → 'Scale for all panels (a-g)*
*Added Scale for the inset map.*

Figure 6: For the 'b' figures, consider changing the Y axis to %'s, as this is how you present the data in the text. This Y axis also needs a label! X axis labels needed for both a and b!
*Added labels to all X and Y axes and changed values to percentage in Panel b.*

Figure 7: Add X axis labels.
*Added X axis labels.*

Table 2: Caption: correct 'ices helves'
*Corrected.*

**Supplementary**

Figure S3: Give the actual image information for imagery used. Apply this where appropriate elsewhere too.
*Added the dates on which the images were captured.*

Figure S6: Legend needed. If you are using REMA data it needs to be detailed in the methods of your paper.
*Added Legend to the figure.*
*Added a sentence at the end of Section 3.1 – 'For topographic assessments we used the 100 m product of Reference Elevation Model of Antarctica (REMA, Howat et al. (2019)).'*

Figure S7: Not referred to in the text? Also this builds heavily on the work of Moussavi et al. (2020), correct? This should be acknowledged. The figure also provides no info on the threshold values used.
*Reference to the figure is added in the rephrased methodology section. Threshold values and legend have been added to the figure. A flow chart of the methodology has also been added as Figure S8.*

A table of the satellite data used and data availability should be included as supplementary info.
*Added ScenesList.csv as additional supplementary document.*

**References**

Dell, R., Arnold, N., Willis, I., Banwell, A. F., Williamson, A., Pritchard, H. D., and Orr, A.: Lateral meltwater transfer across an Antarctic ice shelf, Cryosphere, 14, 2313–2330, https://doi.org/10.5194/tc-14-2313-2020, 2020.

Dell, R., Banwell, A. F., Willis, I. C., Arnold, N. S., Halberstadt, A. R. W., Chudley, T. R., and Pritchard, H. D.: Supervised classification of slush and ponded water on Antarctic ice shelves using Landsat 8 imagery, J. Glaciol., 68, 401–414, https://doi.org/10.1017/jog.2021.114, 2022.

Echelmeyer, K., Clarke, T. S., and Harrison, W. D.: Surficial glaciology of Jakobshavns Isbræ, West Greenland: Part I. Surface morphology, J. Glaciol., 37, 368–382, https://doi.org/10.3189/S0022143000005803, 1991.

Howat, I. M., Porter, C., Smith, B. E., Noh, M.-J., and Morin, P.: The Reference Elevation Model of Antarctica, Cryosph., 13, 665–674, https://doi.org/10.5194/tc-13-665-2019, 2019.

Kingslake, J., Ng, F., and Sole, A.: Modelling channelized surface drainage of supraglacial lakes, J. Glaciol., 61, 185–199, https://doi.org/10.3189/2015JoG14J158, 2015.

Kingslake, J., Ely, J. C., Das, I., and Bell, R. E.: Widespread movement of meltwater onto and across Antarctic ice shelves, Nature, 544, 349–352, https://doi.org/10.1038/nature22049, 2017.

Lenaerts, J. T. M., Lhermitte, S., Drews, R., Ligtenberg, S. R. M., Berger, S., Helm, V., Smeets, C. J. P. P., Broeke, M. R. van den, van de Berg, W. J., van Meijgaard, E., Eijkelboom, M., Eisen, O., and Pattyn, F.: Meltwater produced by wind–albedo interaction stored in an East Antarctic ice shelf, Nat. Clim. Chang., 7, 58–62, https://doi.org/10.1038/nclimate3180, 2017.

Moussavi, M., Pope, A., Halberstadt, A. R. W., Trusel, L. D., Cioffi, L., and Abdalati, W.: Antarctic Supraglacial Lake Detection Using Landsat 8 and Sentinel-2 Imagery: Towards Continental Generation of Lake Volumes, Remote Sens., 12, 134, https://doi.org/10.3390/rs12010134, 2020.

Pope, A., Scambos, T. A., Moussavi, M., Tedesco, M., Willis, M., Shean, D., and Grigsby, S.: Estimating

supraglacial lake depth in West Greenland using Landsat 8 and comparison with other multispectral methods, Cryosphere, 10, 15–27, https://doi.org/10.5194/tc-10-15-2016, 2016.

Stokes, C. R., Sanderson, J. E., Miles, B. W. J., Jamieson, S. S. R., and Leeson, A. A.: Widespread distribution of supraglacial lakes around the margin of the East Antarctic Ice Sheet, Sci. Rep., 9, 1–14, https://doi.org/10.1038/s41598-019-50343-5, 2019.

Williamson, A. G., Banwell, A. F., Willis, I. C., and Arnold, N. S.: Dual-satellite (Sentinel-2 and Landsat 8) remote sensing of supraglacial lakes in Greenland, Cryosphere, 12, 3045–3065, https://doi.org/10.5194/tc-12-3045-2018, 2018.